# Time-Evolving Dynamical System for Learning Latent Representations of Mouse Visual Neural Activity

**Liwei Huang**[1,2], **Zhengyu Ma**[2],[*] **Liutao Yu**[2], **Huihui Zhou**[2], **Yonghong Tian**[1,2,3]

[1]School of Computer Science, Peking University, China
[2]Peng Cheng Laboratory, China
[3]School of Electronic and Computer Engineering, Shenzhen Graduate School, Peking University, China
huanglw20@stu.pku.edu.cn,
{mazhy, yult, zhouhh}@pcl.ac.cn, yhtian@pku.edu.cn

## Abstract

Seeking high-quality representations with latent variable models (LVMs) to reveal the intrinsic correlation between neural activity and behavior or sensory stimuli has attracted much interest. In the study of the biological visual system, naturalistic visual stimuli are inherently high-dimensional and time-dependent, leading to intricate dynamics within visual neural activity. However, most work on LVMs has not explicitly considered neural temporal relationships. To cope with such conditions, we propose Time-Evolving Visual Dynamical System (TE-ViDS), a sequential LVM that decomposes neural activity into low-dimensional latent representations that evolve over time. To better align the model with the characteristics of visual neural activity, we split latent representations into two parts and apply contrastive learning to shape them. Extensive experiments on synthetic datasets and real neural datasets from the mouse visual cortex demonstrate that TE-ViDS achieves the best decoding performance on naturalistic scenes/movies, extracts interpretable latent trajectories that uncover clear underlying neural dynamics, and provides new insights into differences in visual information processing between subjects and between cortical regions. In summary, TE-ViDS is markedly competent in extracting stimulus-relevant embeddings from visual neural activity and contributes to the understanding of visual processing mechanisms. Our codes are available at *https://github.com/Grasshlw/Time-Evolving-Visual-Dynamical-System*.

## 1   Introduction

With the rapid development of neural recording technologies, researchers are able to simultaneously record the spiking activity of large populations of neurons, opening up new avenues for exploring the brain [55]. For high-dimensional neural data analysis, an important scientific problem is how to account for the intrinsic correlation between neural activity and behavioral patterns or sensory stimuli. One influential approach is latent variable models (LVMs), which construct low-dimensional latent representations that bridge to behavior or stimuli and explain neural activity well [47, 5, 56, 26, 34]. To deal with the increasing scale of neural population activity, LVMs have evolved from simple mathematical models early on, such as principal component analysis and factor analysis [9, 61, 44], to complex artificial neural networks. More recently, advanced deep learning algorithms have enabled LVMs to extract high-quality representations from neural activity without knowledge of experimental labels [59, 42, 22, 36], or to incorporate behavioral information into models to constrain the shaping of latent variables [37, 25, 45, 52, 2, 23]. These approaches have contributed to the analysis of neural

---

[*]Corresponding author.

39th Conference on Neural Information Processing Systems (NeurIPS 2025).

activity in various ways, including predicting neural responses [42, 27], decoding related motion patterns or simple visual scenes [36, 48], and constructing interpretable latent structures [63, 3].

Although LVMs have sparked strong interest in uncovering the underlying dynamical structure of neural activity, most studies have focused on neural data recorded from motor brain regions under specific controlled behavioral settings [13, 42, 63, 36, 43], such as pre-planned reaching movements [18]. There is a paucity of studies on LVMs exploring neural data from the visual cortex [21, 62, 48]. Given the significant challenge of elucidating the correlation between neural activity and visual stimuli [15, 28, 58, 17], the development of robust models for extracting vision-related latent representations is critically imperative. Furthermore, due to the complex visual neural dynamics, incorporating temporal structure into models to obtain time-dependent latent representations may be a key point for the analysis of visual neural activity. However, there is also a lack of LVMs that explicitly model neural temporal relationships under visual stimuli.

In this work, we propose the Time-Evolving Visual Dynamical System (TE-ViDS), which aims to generate high-quality latent representations from visual neural activity by disentangling neural components related to visual stimuli from those influenced by internal states. First, we introduce temporal structures to explicitly establish temporal relationships in latent variables, allowing them to evolve over time and capture the temporal dependency inherent in neural activity. Second, to sufficiently utilize the characteristics of visual neural activity, we adopt the split structure approach [36] and design distinct loss functions to construct two specialized parts of latent representations. The *external* latent representations aim to capture stimulus-relevant components within visual neural activity, while the *internal* latent representations reflect the dynamical internal states that influence the animal's sensory capability [39, 4]. In doing so, TE-ViDS can generate latent representations that are more relevant to visual stimuli while also explaining the effects of internal states. We evaluate our model on synthetic and mouse visual datasets, comparing it with leading alternatives. Our model demonstrates its ability to construct meaningful latent representations, yield a greater degree of correlation between neural activity and visual stimuli, and explain the visual information processing mechanisms of the mouse visual cortex. Specifically, our main contributions are as follows.

- We introduce state factors to filter temporal information, enabling TE-ViDS to progressively compress neural activity and evolve latent variables. Regarding the distinct objectives of the two parts of latent representations, we apply self-supervised contrastive learning to shape the *external* and utilize a time-dependent prior distribution to guide the *internal*.

- Through evaluation on synthetic datasets, we show that our model more effectively recovers latent structure and handles time-sequential data well.

- Through evaluation on mouse visual datasets, we demonstrate that our model outperforms alternative models in decoding natural scenes and natural movies, as well as in extracting clear temporal trajectories of neural dynamics at different time scales.

- Further analysis reveals that our model can explain the potential variability in visual information processing between subjects and provide new evidence for the functional hierarchy of the mouse visual cortex.

## 2 Related Work

With the advancement of deep learning, the application of cutting-edge learning algorithms and the innovative design of model structures have greatly promoted the development of LVMs in neuroscience. Some prominent works are summarized below.

The approaches based on neural networks have become major avenues for discovering latent representations underlying neural activity, which better elucidate the mechanisms of neural representations. A well-known model is latent factor analysis via dynamical systems (LFADS), which used RNNs in a sequential VAE framework to extract precise firing rate estimates and predict observed behavior for single-trial data on motor cortical datasets [42, 29, 30]. Recurrent switching linear dynamical systems (rSLDS) [35, 53] and low-rank RNNs [41] also introduced recurrent structures, facilitating the understanding of complex nonlinear neural dynamics. Through specific latent variable designs, pi-VAE [63] and Swap-VAE [36] built interpretable latent structures linked to motor behavior patterns. Furthermore, numerous studies have made significant contributions to achieving the goal of dissociating behaviorally relevant and irrelevant components in neural activity. Targeted neural

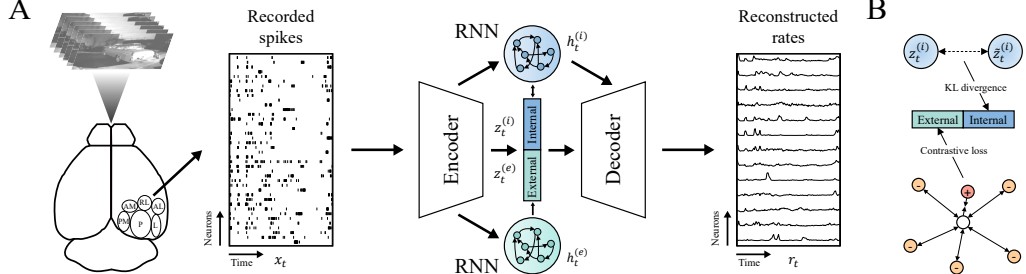

Figure 1: The method overview. **A**. The illustration of TE-ViDS for analyzing visual neural activity in the mouse visual cortex. The encoder extracts spatial features from sequential spike data. The latent variables are evolved conditionally on features of the encoder and RNNs' state factors over time. The decoder maps latent variables to inferred firing rates. **B**. The illustration of different learning objectives for the two parts of latent representations of TE-ViDS. For *external* latent representations, we apply contrastive loss to encourage them to distinguish the stimulus-relevant components. Given a reference sample (white dot), the red dot is a positive sample and the orange dots are negative samples. For *internal* latent representations, we use the KL divergence to constrain their distribution to a time-dependent prior distribution.

dynamical modeling (TNDM), based on LFADS, constructed a two-pathway structure for separation [25]. PSID [45], DFINE [1], and DPAD [46] were dedicated to introducing supervised information into dynamical systems to capture behavior-relevant dynamics.

In the visual domain, several latent variable models have made an effort to advance the understanding of visual neural dynamics. One work used Gaussian process models [19] and found that anesthesia-induced internal state fluctuations lead to correlated variability, while another proposed a network-based linear dynamical system (fLDS) that captures neural variability under drifting grating stimuli [21]. In addition, a rectified latent variable model built latent variables that show a spectrum of functional groups as neurons in the mouse visual cortex under drifting gratings [40], and a flow-based generative model recovered the distribution of visual neural activity [7]. Recently, CEBRA, a self-supervised learning model, obtained consistent latent representations and made progress in decoding movies [48]. Although these studies cover various types of models, most of them are limited to simple visual stimuli and the task of reconstructing neural responses, leaving a gap for constructing high-quality latent representations from visual neural activity under naturalistic stimuli.

## 3 Methods

To generate latent representations that can effectively explain the characteristics of visual neural activity, we introduce the Time-Evolving Visual Dynamical System (TE-ViDS), which evolves two parts of representations over time. In this section, we elaborate the concrete implementations of the model architecture and model learning (visualized in Figure 1).

**Basic notations.** Neural responses recorded from large numbers of neurons are mostly in the form of spikes, which are regarded as discrete events. In practice, it is common to discretize a period of time into small time windows and calculate the number of spikes within each window. Consequently, for the neural activity of a population of neurons over a period of time, we define a sequence input as $\mathbf{x} = (\mathbf{x}_1, \mathbf{x}_2, \dots, \mathbf{x}_T) \in \mathbb{R}^{T \times N}$, which represents spike counts of $N$ neurons within $T$ time windows. The corresponding low-dimensional latent variable is denoted as $\mathbf{z} = (\mathbf{z}_1, \mathbf{z}_2, \dots, \mathbf{z}_T) \in \mathbb{R}^{T \times M}$.

### 3.1 Model Architecture

As aforementioned, TE-ViDS compresses visual neural activity into *external* latent representations and *internal* latent representations ($\mathbf{z}_t = [\mathbf{z}_t^{(e)}, \mathbf{z}_t^{(i)}]$). To evolve the two latent variables, we incorporate two dynamical systems into our model, using the state factor $\mathbf{h}_t$ to sift and accumulate temporal information [8, 20, 12]. In this way, the latent variables of the current time step are only conditioned on the input and state factors of the previous time steps, facilitating the capture of dynamics in causal order.

*External* **latent variables.** This part of latent representations aims to capture the stimulus-relevant components of neural activity. To maintain our focus on variability arising from internal brain states, we construct *external* latent variables as deterministic values:

$$\mathbf{z}_t^{(e)} = f_{\text{enc}}^{(e)}\left(f_{\text{x}}(\mathbf{x}_t), \mathbf{h}_{t-1}^{(e)}\right). \tag{1}$$

*Internal* **latent variables.** This part of latent representations aims to reflect dynamical internal states that contain high variability and noise. To model such dynamics, *internal* latent variables are constructed as stochastic values that evolve over time. The tractable parameterized distribution (approximate posterior) is conditioned on both $\mathbf{x}_t$ and $\mathbf{h}_{t-1}^{(i)}$, while the prior distribution is conditioned solely on $\mathbf{h}_{t-1}^{(i)}$, endowing it with a certain degree of spontaneity:

$$\mathbf{z}_t^{(i)}\Big|\mathbf{x}_{1:t}, \mathbf{h}_{1:t-1}^{(i)} \sim \mathcal{N}(\boldsymbol{\mu}_{z,t}, \boldsymbol{\sigma}_{z,t}^2 \cdot \mathbf{I}), [\boldsymbol{\mu}_{z,t}, \boldsymbol{\sigma}_{z,t}] = f_{\text{enc}}^{(i)}\left(f_{\text{x}}(\mathbf{x}_t), \mathbf{h}_{t-1}^{(i)}\right), \tag{2}$$

$$\tilde{\mathbf{z}}_t^{(i)}\Big|\mathbf{h}_{1:t-1}^{(i)} \sim \mathcal{N}(\tilde{\boldsymbol{\mu}}_{z,t}, \tilde{\boldsymbol{\sigma}}_{z,t}^2 \cdot \mathbf{I}), [\tilde{\boldsymbol{\mu}}_{z,t}, \tilde{\boldsymbol{\sigma}}_{z,t}] = f_{\text{prior}}^{(i)}\left(\mathbf{h}_{t-1}^{(i)}\right). \tag{3}$$

**Neural activity reconstruction.** High-quality latent representations, as compressed forms of the original neural activity, are required to effectively reconstruct their input. In practice, to enrich the dynamic information for a more accurate reconstruction, the spike count observation is related not only to the latent variables but also to the internal state factors:

$$\hat{\mathbf{x}}_t\Big|\mathbf{z}_{1:t}^{(e)}, \mathbf{z}_{1:t}^{(i)}, \mathbf{h}_{1:t-1}^{(i)} \sim P(\mathbf{r}_t), \mathbf{r}_t = f_{\text{dec}}\left(\mathbf{z}_t^{(e)}, \mathbf{z}_t^{(i)}, \mathbf{h}_{t-1}^{(i)}\right), \tag{4}$$

where the parameterized distribution $P$ is chosen to be Poisson [21], i.e., the actual inferred neural activity is spike firing rates $\mathbf{r}_t$.

**Recurrent neural networks.** The state factor is updated by GRU [11]. By selectively integrating and exploiting input and latent variables, the state factor is crucial for capturing complex sequential dynamics. Besides, since the animal's dynamical internal states are inevitably affected by visual stimuli, $\mathbf{h}_t^{(e)}$ and $\mathbf{h}_t^{(i)}$ corresponding to the two latent representations are updated differently:

$$\begin{aligned}
\mathbf{h}_t^{(e)} &= f_{\text{GRU}}^{(e)}\left(f_{\text{x}}(\mathbf{x}_t), \mathbf{h}_{t-1}^{(e)}\right), \\
\mathbf{h}_t^{(i)} &= f_{\text{GRU}}^{(i)}\left(f_{\text{x}}(\mathbf{x}_t), \mathbf{z}_t^{(e)}, \mathbf{z}_t^{(i)}, \mathbf{h}_{t-1}^{(i)}\right).
\end{aligned} \tag{5}$$

All the functions $f$ above are parameter-learnable neural networks (see Appendix A for details).

### 3.2 Model Learning

Given that the two parts of latent representations hold different objectives for disentanglement, we introduce the following loss function to optimize all learnable parameters in an end-to-end manner:

$$\mathcal{L} = \mathcal{L}_{\text{recons}} + \beta \mathcal{L}_{\text{contrastive}} + \gamma \mathcal{L}_{\text{regular}}. \tag{6}$$

The first term which deals with the objective of inferring neural activity is formulated as $\frac{1}{T}\sum_{t=1}^{T}\left[\mathcal{L}_{\text{P}}(\mathbf{x}_t, \mathbf{r}_t)\right]$, where $\mathcal{L}_{\text{P}}$ is Poisson negative log likelihood.

The second term encourages *external* latent representations to distinguish stimulus-relevant components through self-supervised contrastive learning. Specifically, for a given sample $\mathbf{x} = (\mathbf{x}_1, \ldots, \mathbf{x}_T)$, we randomly select another sequence offset by several time steps as a positive sample, denoted $\mathbf{x}_{\text{pos}} = (\mathbf{x}_{1+\Delta}, \ldots, \mathbf{x}_{T+\Delta})$, where $\Delta$ can be positive or negative. The offset is smaller than the length of the sequence to ensure that the positive sample pairs overlap, i.e., the visual stimuli corresponding to the positive sample pairs are similar, which establishes an association between the *external* latent variables and the visual stimuli. Then, a mini-batch of negative samples is randomly selected from the entire training set. Finally, we compute the *external* latent variables of all selected samples and apply the NT-Xent loss [10] as $\mathcal{L}_{\text{contrastive}}$ to them. In doing so, *external* latent representations of the positive sample pairs are driven to be distinguished from the negative samples. For this term, we do not apply the time-wise operation, but flatten the temporal and spatial dimensions of the *external* latent variables for the loss computation. In addition, to enhance the effect of the positive sample, we adopt the practice of the swap operation [36]. We exchange the *external* latent variables

of the given sample with those of the positive sample while keeping the *internal* latent variables unchanged. The new latent representations are then used to compute new inferred firing rates and an additional reconstruction loss.

The third term measures the difference between the prior and the approximate posterior of the *internal* latent variables by the KL divergence, formulated as $\frac{1}{T} \sum_{t=1}^{T} \left[ \mathrm{D}_{\mathrm{KL}}(\mathbf{z}_t^{(i)} \| \tilde{\mathbf{z}}_t^{(i)}) \right]$. Besides, we compute the L2 norm of the expectation and log-variance of the prior distribution as a regularization to avoid excessive fluctuations over time and to stabilize model training.

$\beta$ and $\gamma$ are hyperparameters used to control the severity of the penalty for each loss term. The entire loss function actually obeys the strategy of maximizing the evidence lower bound (ELBO) of the marginal log likelihood in the VAE framework [31], while also accounting for the impact of introducing the contrastive loss [57]. A detailed derivation is given in Appendix B.

# 4 Experiments

To evaluate the utility of the latent representations constructed by TE-ViDS for the analysis of visual neural activity, we perform a series of experiments to compare our model with alternative LVMs from two aspects, similar to studies focusing on motor brain regions [36, 48]. First, we quantify the performance in decoding visual stimuli using latent representations, which has long served as a research hotspot for unraveling the mechanisms of visual processing [28, 58]. Second, we assess the clarity of latent temporal trajectories extracted from neural dynamics.

## 4.1 Datasets

We carry out the experiments on two commonly used synthetic datasets and a well-known publicly available mouse visual neural dataset.

**Synthetic datasets.** The two synthetic datasets are crucial for evaluating the fundamental ability of LVMs to construct accurate latent variables. One is a non-temporal dataset generated from several sets of labels [63, 36], facilitating the assessment of class discrimination capabilities. The other is a temporal dataset constructed by the Lorenz system to test for extracting temporal relationships [21, 54]. A detailed description of the generating procedure is presented in Appendix C.

**Mouse visual neural dataset.** We use a subset of the Allen Brain Observatory Visual Coding dataset [51], which has been used in a variety of studies, including the construction of brain-like networks [50], the modeling of functional mechanisms [6, 14], and the decoding of visual stimuli [48]. This dataset was collected by Neuropixel probes from six mouse visual cortical regions simultaneously, including VISp, VISl, VISrl, VISal, VISpm, and VISam. Notably, neural activity was recorded while mice passively viewed visual stimuli without any task-driven behavior. The dataset comprises 32 sessions, each involving one mouse. In this work, we choose to analyze five mice (subjects) that have the maximum number of recorded neurons (see Appendix D), and these neurons are evenly distributed across all regions (the coefficient of variation for the number of neurons across six regions is below 0.5). We focus on neural activity in response to natural scenes and a natural movie. For natural scenes, 118 images are presented in random order, each for 250ms and 50 trials. The neural activity in the form of spike counts is binned into 10-ms windows so that each trial contains 25 time points. The natural movie is 30 seconds long with a frame rate of 30Hz, presented for 10 trials. We bin the spike counts with a sampling frequency of 120Hz and align them with the movie timestamps, resulting in four time points for each frame.

Since spike responses are quite variable across trials even under identical experimental conditions, we conduct the basic evaluation on "held-out" trials. Specifically, for each dataset, we randomly split all trials into 80% for training, 10% for validation, and 10% for test.

## 4.2 Evaluation Metric

Following previous studies [54, 48], we apply two quantitative metrics for evaluation.

**Reconstruction score.** For the two synthetic datasets, we use linear regression to fit the latent variables of models to the true latent variables, and report the $R^2$ as the reconstruction score.

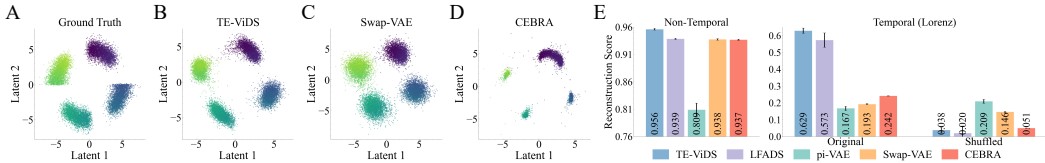

Figure 2: Results on synthetic datasets. **A**. The true latent variables of the non-temporal dataset. **B-D**. The inferred latent variables of our model and some alternative models. **E**. The reconstruction scores of all models on the non-temporal and temporal datasets. The standard error is computed on 10 runs with different random initializations.

**Decoding score.** For the mouse visual neural dataset, we quantify the performance of models in decoding natural scenes/natural movie frames. In terms of natural scene decoding, we obtain the latent variables of the last 20 time points (50ms-250ms) in each trial, since there is a response latency in the mouse visual cortex when presented with static stimuli [51]. These latent variables are then concatenated into a vector to form the latent representations of each trial, thereby retaining maximal temporal information. We use the KNN algorithm to classify the latent representations of each trial, i.e., to decode the corresponding natural scenes. In terms of natural movie frame decoding, we compute the latent variables of four time points within each frame, averaging them to create the latent representations for that frame. We also use the KNN to predict movie frames based on latent representations (900 frames in total, i.e., 900 classes). The specific KNN procedure is as follows: First, we fit the KNN on the training set and search for the optimal hyperparameter (the number of neighbors) on the validation set, using classification accuracy as the metric. The search range is set to odd numbers between 1 and 20, with odd numbers chosen to reduce decision uncertainty. Then, the accuracy on the test set is reported as the decoding score. Notably, for movie frame decoding, we follow the established approach of CEBRA [48], considering an error of less than 1s (constraint window) between the predicted frame and the true frame as a correct prediction.

### 4.3 Alternative Models

For a comprehensive analysis, we compare TE-ViDS with several leading LVMs, including three generative models (a sequential: LFADS [42], a supervised: pi-VAE [63], and a self-supervised: Swap-VAE [36]) and a nonlinear encoding method with contrastive learning (CEBRA) [48]. Specifically, pi-VAE[2] incorporates additional supervised information for latent variable construction, but does not model temporal dynamics. Swap-VAE utilizes contrastive learning to construct separated latent variables, aligning with our disentanglement goal, but it also does not explicitly model temporal dynamics. For the two models that are able to model the temporal relationships, LFADS processes sequential neural activity using bidirectional RNNs, while CEBRA encodes temporal features of sequential data with fixed convolutional kernels. Furthermore, for fair comparisons, we build a small version of our model (TE-ViDS-small) with fewer trainable parameters than Swap-VAE (see Appendix E for the number of trainable parameters).

For each dataset, all models are set to latent variables of the same dimension and trained for more than 20,000 iterations until convergence. Additionally, we apply the hyperparameter grid search for all models. More details of the training setup are presented in Appendix F.

### 4.4 Results on Synthetic Datasets

First, we visualize the reconstructed latent variables on the non-temporal dataset (Figure 2A-D). TE-ViDS reliably separates the different clusters as well as recovers the structure of the true latent variables to form clear arcs. In contrast, some of the alternative models fail to construct similar structures despite separating clusters (Swap-VAE and CEBRA), and others even struggle to separate four clusters (LFADS and pi-VAE; Appendix G). Quantitatively, the reconstruction scores also suggest that our model outperforms all alternative models (Figure 2E).

For the original temporal dataset, TE-ViDS performs significantly better than those models that process sequential spikes at each time point independently, and moderately better than LFADS

---

[2]pi-VAE incorporates the label prior during training, but the inferred latent variables are built without the label prior at the evaluation stage. This way is used in all experiments.

Table 1: The decoding scores (%) for 118 natural scenes on the mouse visual neural dataset. The standard error is computed on 10 runs with different random initializations.

| Models | Mouse 1 | Mouse 2 | Mouse 3 | Mouse 4 | Mouse 5 |
|---|---|---|---|---|---|
| PCA | 0.59 | 1.53 | 1.53 | 0.80 | 0.85 |
| LFADS | 30.76±0.65 | 16.46±0.49 | 22.20±0.26 | 19.69±0.38 | 4.69±0.22 |
| pi-VAE | 7.49±0.77 | 19.42±0.62 | 22.92±0.37 | 13.71±1.39 | 2.22±0.64 |
| Swap-VAE | 32.81±1.47 | 24.34±0.57 | 14.36±1.31 | 14.85±1.29 | 3.92±0.37 |
| CEBRA | 1.53±0.15 | 3.42±0.07 | 4.86±0.19 | 2.81±0.23 | 1.08±0.10 |
| **TE-ViDS-small** | 47.08±2.86 | 23.95±1.39 | 29.08±1.47 | 34.95±0.90 | **9.93±0.27** |
| **TE-ViDS** | **50.86±0.81** | **27.24±0.47** | **29.90±0.43** | **38.05±0.53** | 9.44±0.20 |

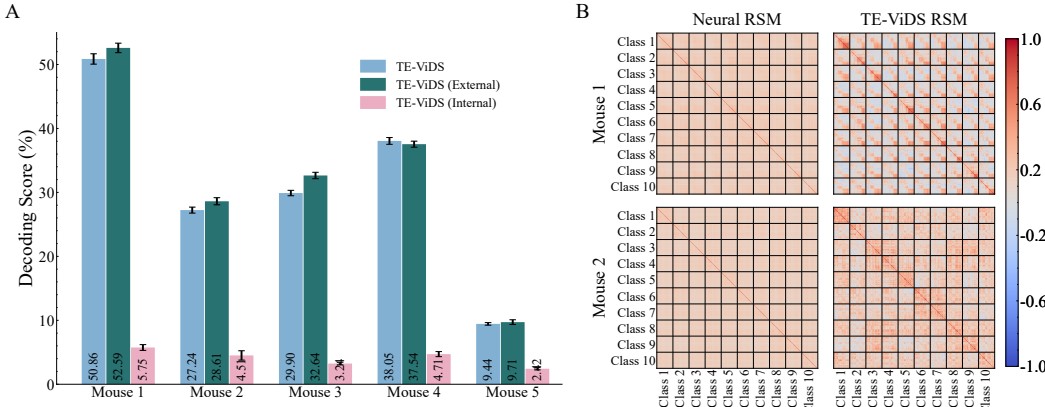

Figure 3: Results on the mouse neural dataset under natural scene stimuli. **A**. The decoding scores (%) of the *full*, *external* and *internal* latent representations of TE-ViDS for 118 natural scenes. **B**. RSMs computed on the original neural representations and TE-ViDS's latent representations, respectively (Mouse 1 and Mouse 2). Each element in a matrix is the similarity between two trials' representations. Each small square involves comparisons between two natural scenes, containing 50 trials.

which also uses RNNs to process sequential data (Figure 2E). Moreover, when we shuffle the time dimension for each trial data in the original dataset to obtain a dataset without temporal dependency, the performance of our model and LFADS shows a drastic degradation. However, for models utilizing time-jittered positive samples, the performance reduction observed in Swap-VAE and CEBRA was comparatively less severe. These results demonstrate the superiority of our model in dealing with temporal data and its sensitivity to temporal relationships.

### 4.5 Results on the Mouse Neural Dataset under Natural Scene Stimuli

As shown in Table 1, TE-ViDS achieves the highest decoding scores for all mice with a noticeable improvement over other models. In particular, CEBRA, which can encode temporal relationships, instead performs poorly in this downstream task, suggesting that using time filters to extract temporal neural features is less suitable in this case. In contrast, the time-evolving latent representations constructed by our model reliably capture temporal relationships encoded in the mouse visual cortex under different static scene stimuli, which may be helpful for natural scene discrimination. We further evaluate the decoding performance of the *external* and *internal* latent variables separately (Figure 3A). We find that the *external* latent representations significantly outperform the *internal* latent representations, supporting our assumption that the former are tuned to stimulus-relevant components of neural activity, while the latter point to stimulus-irrelevant internal states.

Although our model achieves the highest performance, we find that the decoding scores for different mice show large differences. We further apply Representational Similarity Analysis (RSA) [33, 32] to the original neural activity and TE-ViDS's latent representations. Specifically, we calculate the representational similarity between the representations to each pair of trials using the Pearson correlation coefficient, yielding representational similarity matrices (RSMs). We select ten scenes that elicit the strongest average responses for visualization. By comparing the RSMs of the original

Table 2: The decoding scores (%, in 1s window) for natural movie frames on the mouse visual neural dataset. The standard error is computed on 10 runs with different random initializations.

| Models | Mouse 1 | Mouse 2 | Mouse 3 | Mouse 4 | Mouse 5 |
|---|---|---|---|---|---|
| PCA (baseline) | 8.44 | 28.77 | 25.42 | 21.56 | 11.69 |
| LFADS | 8.94±0.25 | 26.57±2.46 | 26.77±2.23 | 24.76±1.80 | 12.69±1.38 |
| pi-VAE | 10.24±0.31 | 42.51±0.65 | 36.96±0.60 | 38.31±0.52 | 18.08±0.59 |
| Swap-VAE | 12.19±0.20 | 51.31±0.73 | 45.96±0.34 | 41.53±0.63 | 22.70±0.42 |
| CEBRA | 10.62±0.18 | 52.76±0.89 | **61.01±0.76** | 42.11±0.73 | 22.33±0.31 |
| **TE-ViDS-small** | 13.23±0.25 | 64.09±0.41 | 59.36±0.30 | 53.46±0.58 | 29.60±0.26 |
| **TE-ViDS** | **13.88±0.19** | **65.38±0.36** | 59.88±0.72 | **54.33±0.54** | **30.18±0.40** |

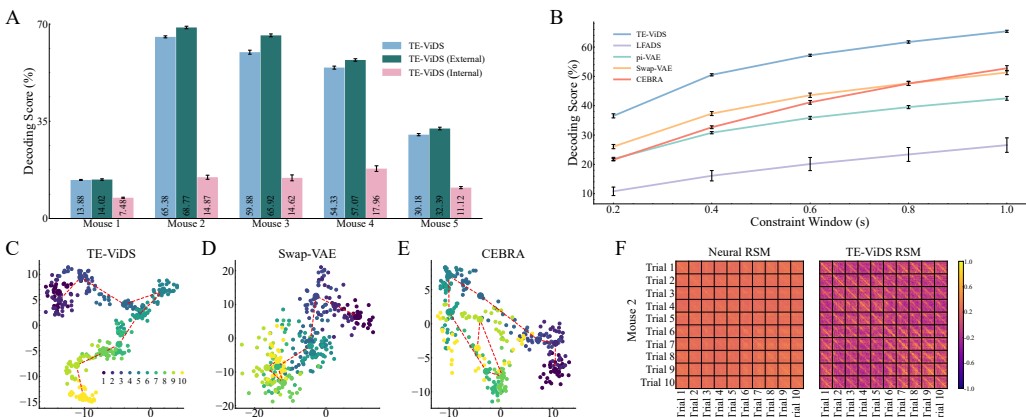

Figure 4: Results on the mouse neural dataset under natural movie stimuli. **A**. The decoding scores (%) of the *full*, *external* and *internal* latent representations of TE-ViDS. **B**. The decoding scores (%) for movie frames under different constraint windows of predicted frames and the true frames. **C-E**. Visualization results of latent trajectories (Mouse 2). Each color corresponds to all frames within 1s. Small dots denote one frame. Large dots denote the average among a group of frames. The red dashed line connects all averages. **F**. RSMs computed on the original neural representations and TE-ViDS's latent representations (Mouse 2). Each element is the similarity between two frames' representations. Each small square involves comparisons between two trials, containing 900 frames.

neural activity and TE-ViDS (Figure 3B), we show that our model can extract clearer intrinsic patterns of neural activity and reduce the impact of useless noise, facilitating the unraveling of information processing mechanisms in the visual cortex. Moreover, we observe that the RSMs of TE-ViDS between two mice exhibit significantly different patterns. For Mouse 1, there are two redder blocks, top left and bottom right, in each of the small squares, suggesting that the neural representations of closer trials in time are more similar and divided into two periods, even under different visual stimuli. This phenomenon may be due to differences in the internal states of this mouse during the two periods, and the effect of such states on neural activity is even stronger than that of visual stimuli to some extent. For Mouse 2, there is no such clear change in state. These results echo some previous studies, showing that sensory performance in mice is strongly influenced by their internal state [4], which may also be an important reason for the variability in neural activity between subjects.

In addition to tasks on "held-out" trials, we perform a more challenging task on "held-out" stimuli. All models are trained with neural responses to 95 natural scenes (approximately 80% of all scenes), and their decoding performance is evaluated on the remaining 23 scenes. The results (see Appendix H) show that TE-ViDS outperforms other models, even for these unseen stimuli.

In summary, these results demonstrate the advantages of our model in decoding natural scene stimuli and extracting fine neural dynamics (see latent trajectories over time in Appendix I). The latent representations constructed by our model help to explain potential differences in visual information processing between individuals, which deserves further exploration.

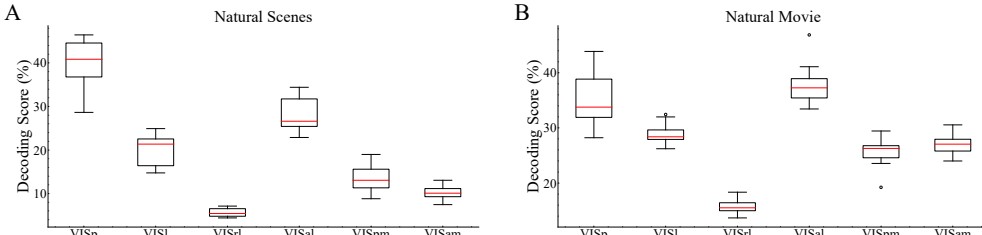

Figure 5: The decoding scores (%) of TE-ViDS for natural scenes/natural movie frames on six mouse visual cortical region datasets. The box plots are based on 10 runs with different random initializations.

## 4.6    Results on the Mouse Neural Dataset under Natural Movie Stimuli

TE-ViDS performs best for four mice (Table 2). Specifically, our model gains more advantage over LFADS than over the others. We also show that the *external* latent representations are more stimulus-relevant (Figure 4A). The results of Figure 4B show that TE-ViDS consistently outperforms alternatives across a broad range of constraint windows, especially in finer-grained frame decoding.

For visualization, we reduce the dimensions of the latent representations of each movie frame using tSNE. We set all frames within 1s as a group and show the trajectories of latent representations for the middle 10s of the movie (Figures 4C-E. We focus on Mouse 2, for which most models achieve the highest scores. See Appendix I for the results of the other parts of the movie and the other models). Compared to alternative models, the representations of TE-ViDS show clear temporal structures along movie frames with less overlap and entanglement between different time groups.

For further analysis, we also compute RSMs of the original neural activity and TE-ViDS on Mouse 2 (Figure 4F). Different from the RSMs in Figure 3B, each element in these matrices is the representational similarity between the representations to each pair of movie frames and each small square involves comparisons between two trials. The comparison between neural RSM and TE-ViDS RSM provides the same evidence that our model extracts clearer patterns and reduces noise. In each small square of TE-ViDS RSM, brighter regions are near the diagonal, showing that the neural representations of neighboring frames are more similar, even for different trials. This pattern suggests that under continuous and long-duration visual stimuli, this mouse has a stronger neural response to stimuli and is less affected by its internal state. This may reveal differences in the mouse's visual encoding ability under short-duration static stimuli and long-duration continuous stimuli.

To conclude, our model achieves the highest decoding scores for natural movie frames. Furthermore, we also perform experiments on "held-out" movie frame stimuli, in which our model performs best on decoding movie frames (Appendix H). Importantly, our model constructs meaningful latent representations related to the content and temporal structure of movie stimuli at large time scales, providing insights into visual information processing mechanisms for long-duration stimuli.

## 4.7    Experiments for Mouse Cortical Regions

To explore the ability of six mouse visual cortical regions to process visual stimuli, we randomly sampled 150 neurons from each of the six regions of five selected mice to construct visual neural activity datasets, respectively. We train TE-ViDS on these six datasets and evaluate its decoding performance for natural scenes/natural movie frames. The experiment is repeated ten times.

As shown in Figure 5, the decoding performance varies considerably across regions and the performance trends across regions between decoding the two types of stimuli are essentially the same. The anatomical hierarchy of the mouse visual cortex [51] shows that VISp is the primary region, while VISl, VISrl and VISal are the mid-level regions, and VISpm and VISam are the high-level regions. Additionally, VISrl is a multi-sensory region [14] that receives multi-modal sensory input. Our results show that the decoding performance is higher in the primary and mid-level regions than in the high-level regions, with the lowest performance observed in VISrl. First, the poor performance of VISrl may be due to the fact that, as a multi-sensory region, it is not well driven by visual stimuli alone. Second, while some previous studies have suggested that the mouse visual cortex is organized in a parallel structure [49, 24], the differences in decoding performance across regions may provide

evidence that there is heterogeneous and specialized visual encoding information for each brain region. These results echo the anatomical work [51] to some extent and provide new insights into the functional hierarchy of the mouse visual cortex.

## 5 Discussion

This work presents a novel latent variable model, TE-ViDS, by introducing temporal structures to explicitly establish temporal relationships and constructing two parts of latent representations to disentangle the stimulus-relevant components from visual neural activity. The results of synthetic and mouse neural datasets demonstrate that our model outperforms alternative models and builds latent representations that are strongly correlated with visual input, revealing intrinsic correlations between visual neural activity and visual stimuli. A series of ablation studies demonstrates the effectiveness of our model's components (Appendix J). Furthermore, TE-ViDS aids in explaining the variability in visual information processing between subjects and provides computational evidence for the functional hierarchy of the mouse visual cortex, which may contribute to the computational neuroscience community.

There is a paucity of research analyzing visual neural activity with LVMs. CEBRA has built consistent latent representations from multimodal visual neural activity [48]. Our work takes a further step in the development of powerful LVMs to yield stimulus-correlated latent representations and capture neural dynamics. However, there are still some limitations. First, in the absence of recorded internal states or behavioral information, we are unable to quantitatively assess the interpretability of *internal* latent variables and find it difficult to interpret the functional role of inferred *internal* latent representations. Second, our model exhibits substantial variability in decoding performance across individual mice. Subsequent analyses suggest that the variability in neural activity between subjects and trials [60, 38] may be caused by the animal's internal states. This could represent a potential breakthrough for LVMs to consistently construct high-quality stimulus-correlated latent representations across conditions, but further exploration is required.

Last but not least, our approach is not limited to studying visual neural spikes from mice and can be extended to neural data from other species and modalities to investigate broader principles of biological coding mechanisms, facilitating the development of computational neuroscience.

## Acknowledgments and Disclosure of Funding

This work is supported by grants from the Shenzhen KQTD (No. 20240729102051063) and the National Natural Science Foundation of China (No. 62027804, No. 62425101, No. 62332002, No. 62088102, and No. 62206141).

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

## A  Detailed Structure of TE-ViDS

The encoder and decoder of TE-ViDS are derived from Swap-VAE. The encoder consists of three blocks, the first two of which are sequentially stacked with a linear layer, a batch normalization, and a ReLU activation. The last block differs from Swap-VAE in that it additionally introduces the hidden states of the GRU as input. The output dimensions of the three blocks are $N$, $M$, and $M$, where $N$ is the number of neurons and $M$ is the number of latent variables. The decoder is a symmetric structure of the encoder, where the first two blocks are also sequentially stacked with a linear layer, a batch normalization, and a ReLU activation, and the last block is a linear layer followed by a SoftPlus activation. We set the dimensions of the two latent variables to be equal and use a one-layer GRU for each of them, where the dimensions of the hidden states are equal to the dimensions of the latent variables. The operations of each module are shown in Figure 6.

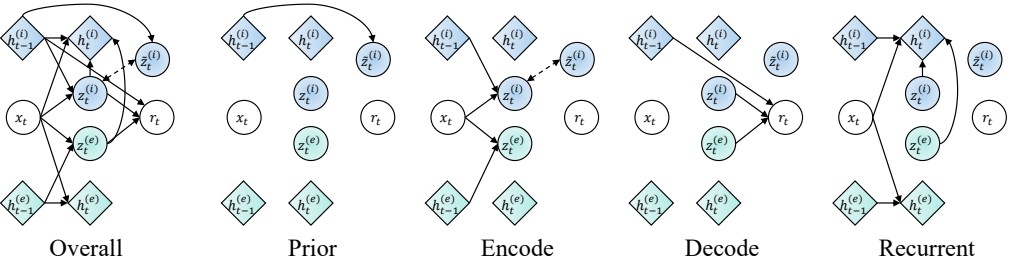

Figure 6: The operations of each module in TE-ViDS.

## B  Derivation of the Loss Function of TE-ViDS

The loss function of TE-ViDS obeys the strategy to maximize the likelihood of the joint sequential distribution $p(\mathbf{x}_{1:T})$. Involving the latent variables $\mathbf{z}_{1:T}$, we have the variational lower bound:

$$
\begin{aligned}
\log p(\mathbf{x}_{1:T}) &= \int q(\mathbf{z}_{1:T}|\mathbf{x}_{1:T}) \log p(\mathbf{x}_{1:T}) d\mathbf{z}_{1:T} \\
&= \int q(\mathbf{z}_{1:T}|\mathbf{x}_{1:T}) \log \frac{p(\mathbf{x}_{1:T}, \mathbf{z}_{1:T})}{p(\mathbf{z}_{1:T}|\mathbf{x}_{1:T})} d\mathbf{z}_{1:T} \\
&= \int q(\mathbf{z}_{1:T}|\mathbf{x}_{1:T}) \log \frac{q(\mathbf{z}_{1:T}|\mathbf{x}_{1:T})}{p(\mathbf{z}_{1:T}|\mathbf{x}_{1:T})} d\mathbf{z}_{1:T} + \int q(\mathbf{z}_{1:T}|\mathbf{x}_{1:T}) \log \frac{p(\mathbf{x}_{1:T}, \mathbf{z}_{1:T})}{q(\mathbf{z}_{1:T}|\mathbf{x}_{1:T})} d\mathbf{z}_{1:T} \\
&= \mathrm{KL}(q(\mathbf{z}_{1:T}|\mathbf{x}_{1:T})\|p(\mathbf{z}_{1:T}|\mathbf{x}_{1:T})) + \int q(\mathbf{z}_{1:T}|\mathbf{x}_{1:T}) \log \frac{p(\mathbf{x}_{1:T}, \mathbf{z}_{1:T})}{q(\mathbf{z}_{1:T}|\mathbf{x}_{1:T})} d\mathbf{z}_{1:T} \\
&\geq \int q(\mathbf{z}_{1:T}|\mathbf{x}_{1:T}) \log \frac{p(\mathbf{x}_{1:T}, \mathbf{z}_{1:T})}{q(\mathbf{z}_{1:T}|\mathbf{x}_{1:T})} d\mathbf{z}_{1:T}, \\
&\geq \int q(\mathbf{z}_{1:T}|\mathbf{x}_{1:T}) \log \frac{p(\mathbf{x}_{1:T}, \mathbf{z}_{1:T})}{q(\mathbf{z}_{1:T}|\mathbf{x}_{1:T})} d\mathbf{z}_{1:T} - \mathcal{L}_{\mathrm{contrast}},
\end{aligned}
\tag{7}
$$

where $p(\mathbf{x}_{1:T}, \mathbf{z}_{1:T})$ is the joint distribution as well as $p(\mathbf{z}_{1:T}|\mathbf{x}_{1:T})$ and $q(\mathbf{z}_{1:T}|\mathbf{x}_{1:T})$ is the true posterior and the variational approximate posterior, respectively. The true posterior is intractable.

Considering Equations 1 an 5, we know that $\mathbf{z}_t^{(e)}$ is deterministic values and $\mathbf{h}_t^{(i)}$ is a function of $\mathbf{x}_{1:t}$ and $\mathbf{z}_{1:t}^{(i)}$. Therefore, we have the factorization:

$$
p(\mathbf{x}_{1:T}, \mathbf{z}_{1:T}) = \prod_{t=1}^{T} p(\mathbf{x}_t|\mathbf{z}_{1:t}^{(i)}, \mathbf{z}_{1:t}^{(e)}, \mathbf{x}_{1:t-1}) p(\mathbf{z}_t^{(i)}|\mathbf{x}_{1:t-1}, \mathbf{z}_{1:t-1}^{(i)}),
\tag{8}
$$

$$
q(\mathbf{z}_{1:T}|\mathbf{x}_{1:T}) = \prod_{t=1}^{T} q(\mathbf{z}_t^{(i)}|\mathbf{x}_{1:t}, \mathbf{z}_{1:t-1}^{(i)}),
\tag{9}
$$

where $q(\mathbf{z}_t^{(i)}|\mathbf{x}_{1:t}, \mathbf{z}_{1:t-1}^{(i)})$, $p(\mathbf{z}_t^{(i)}|\mathbf{x}_{1:t-1}, \mathbf{z}_{1:t-1}^{(i)})$ and $p(\mathbf{x}_t|\mathbf{z}_{1:t}^{(i)}, \mathbf{z}_{1:t}^{(e)}, \mathbf{x}_{1:t-1})$ are the distributions defined by Equations 2, 3 and 4, respectively. Based on the above factorization, we decompose the variational lower bound as:

$$
\int q(\mathbf{z}_{1:T}|\mathbf{x}_{1:T}) \log \frac{p(\mathbf{x}_{1:T}, \mathbf{z}_{1:T})}{q(\mathbf{z}_{1:T}|\mathbf{x}_{1:T})} d\mathbf{z}_{1:T}
$$
$$
= \int q(\mathbf{z}_{1:T}|\mathbf{x}_{1:T}) \sum_{t=1}^{T} \left( \log \frac{p(\mathbf{x}_t|\mathbf{z}_{1:t}^{(i)}, \mathbf{z}_{1:t}^{(e)}, \mathbf{x}_{1:t-1}) p(\mathbf{z}_t^{(i)}|\mathbf{x}_{1:t-1}, \mathbf{z}_{1:t-1}^{(i)})}{q(\mathbf{z}_t^{(i)}|\mathbf{x}_{1:t}, \mathbf{z}_{1:t-1}^{(i)})} \right) d\mathbf{z}_{1:T} \qquad (10)
$$
$$
= \sum_{t=1}^{T} \left( \int q(\mathbf{z}_{1:T}|\mathbf{x}_{1:T}) \log \frac{p(\mathbf{x}_t|\mathbf{z}_{1:t}^{(i)}, \mathbf{z}_{1:t}^{(e)}, \mathbf{x}_{1:t-1}) p(\mathbf{z}_t^{(i)}|\mathbf{x}_{1:t-1}, \mathbf{z}_{1:t-1}^{(i)})}{q(\mathbf{z}_t^{(i)}|\mathbf{x}_{1:t}, \mathbf{z}_{1:t-1}^{(i)})} d\mathbf{z}_{1:T} \right).
$$

When we simplify the above log-likelihood to a function $g(\mathbf{x}_{1:t}, \mathbf{z}_{1:t})$, we have:

$$
\int q(\mathbf{z}_{1:T}|\mathbf{x}_{1:T}) g(\mathbf{x}_{1:t}, \mathbf{z}_{1:t}) d\mathbf{z}_{1:T}
$$
$$
= \int \left( \int q(\mathbf{z}_{1:T-1}|\mathbf{x}_{1:T-1}) q(\mathbf{z}_T^{(i)}|\mathbf{x}_{1:T}, \mathbf{z}_{1:T-1}^{(i)}) g(\mathbf{x}_{1:t}, \mathbf{z}_{1:t}) d\mathbf{z}_T \right) d\mathbf{z}_{1:T-1}
$$
$$
= \int \left( q(\mathbf{z}_{1:T-1}|\mathbf{x}_{1:T-1}) g(\mathbf{x}_{1:t}, \mathbf{z}_{1:t}) \int q(\mathbf{z}_T^{(i)}|\mathbf{x}_{1:T}, \mathbf{z}_{1:T-1}^{(i)}) d\mathbf{z}_T \right) d\mathbf{z}_{1:T-1} \qquad (11)
$$
$$
= \int q(\mathbf{z}_{1:T-1}|\mathbf{x}_{1:T-1}) g(\mathbf{x}_{1:t}, \mathbf{z}_{1:t}) d\mathbf{z}_{1:T-1}
$$
$$
= \cdots = \int q(\mathbf{z}_{1:t}|\mathbf{x}_{1:t}) g(\mathbf{x}_{1:t}, \mathbf{z}_{1:t}) d\mathbf{z}_{1:t}.
$$

Therefore, we further decompose Equation 10 as:

$$
\int q(\mathbf{z}_{1:T}|\mathbf{x}_{1:T}) \log \frac{p(\mathbf{x}_{1:T}, \mathbf{z}_{1:T})}{q(\mathbf{z}_{1:T}|\mathbf{x}_{1:T})} d\mathbf{z}_{1:T}
$$
$$
= \sum_{t=1}^{T} \left( \int q(\mathbf{z}_{1:t}|\mathbf{x}_{1:t}) \log \frac{p(\mathbf{x}_t|\mathbf{z}_{1:t}^{(i)}, \mathbf{z}_{1:t}^{(e)}, \mathbf{x}_{1:t-1}) p(\mathbf{z}_t^{(i)}|\mathbf{x}_{1:t-1}, \mathbf{z}_{1:t-1}^{(i)})}{q(\mathbf{z}_t^{(i)}|\mathbf{x}_{1:t}, \mathbf{z}_{1:t-1}^{(i)})} d\mathbf{z}_{1:t} \right)
$$
$$
= \sum_{t=1}^{T} \left( \int q(\mathbf{z}_{1:t}|\mathbf{x}_{1:t}) \log p(\mathbf{x}_t|\mathbf{z}_{1:t}^{(i)}, \mathbf{z}_{1:t}^{(e)}, \mathbf{x}_{1:t-1}) d\mathbf{z}_{1:t} + \right.
$$
$$
\left. \int q(\mathbf{z}_{1:t}|\mathbf{x}_{1:t}) \log \frac{p(\mathbf{z}_t^{(i)}|\mathbf{x}_{1:t-1}, \mathbf{z}_{1:t-1}^{(i)})}{q(\mathbf{z}_t^{(i)}|\mathbf{x}_{1:t}, \mathbf{z}_{1:t-1}^{(i)})} d\mathbf{z}_{1:t} \right)
$$
$$
= \sum_{t=1}^{T} \left( \int q(\mathbf{z}_{1:t}|\mathbf{x}_{1:t}) \log p(\mathbf{x}_t|\mathbf{z}_{1:t}^{(i)}, \mathbf{z}_{1:t}^{(e)}, \mathbf{x}_{1:t-1}) d\mathbf{z}_{1:t} - \right. \qquad (12)
$$
$$
\left. \int q(\mathbf{z}_{1:t-1}|\mathbf{x}_{1:t-1}) \mathrm{KL}(q(\mathbf{z}_t^{(i)}|\mathbf{x}_{1:t}, \mathbf{z}_{1:t-1}^{(i)})\|p(\mathbf{z}_t^{(i)}|\mathbf{x}_{1:t-1}, \mathbf{z}_{1:t-1}^{(i)})) d\mathbf{z}_{1:t-1} \right)
$$
$$
= \int q(\mathbf{z}_{1:T}|\mathbf{x}_{1:T}) \sum_{t=1}^{T} \left( \log p(\mathbf{x}_t|\mathbf{z}_{1:t}^{(i)}, \mathbf{z}_{1:t}^{(e)}, \mathbf{x}_{1:t-1}) - \right.
$$
$$
\left. \mathrm{KL}(q(\mathbf{z}_t^{(i)}|\mathbf{x}_{1:t}, \mathbf{z}_{1:t-1}^{(i)})\|p(\mathbf{z}_t^{(i)}|\mathbf{x}_{1:t-1}, \mathbf{z}_{1:t-1}^{(i)})) \right) d\mathbf{z}_{1:T}
$$
$$
= \mathbb{E}_{q(\mathbf{z}_{1:T}|\mathbf{x}_{1:T})} \left[ \sum_{t=1}^{T} \log p(\mathbf{x}_t|\mathbf{z}_{1:t}^{(i)}, \mathbf{z}_{1:t}^{(e)}, \mathbf{x}_{1:t-1}) - \right.
$$
$$
\left. \mathrm{KL}(q(\mathbf{z}_t^{(i)}|\mathbf{x}_{1:t}, \mathbf{z}_{1:t-1}^{(i)})\|p(\mathbf{z}_t^{(i)}|\mathbf{x}_{1:t-1}, \mathbf{z}_{1:t-1}^{(i)})) \right].
$$

Finally, for a given sequential data x, we have the loss function:

$$\mathcal{L} \simeq \sum_{t=1}^{T} \left( \underbrace{-\log p(\mathbf{x}_t | \mathbf{z}_{1:t}^{(i)}, \mathbf{z}_{1:t}^{(e)}, \mathbf{x}_{1:t-1})}_{\text{reconstruction loss}} + \underbrace{\text{KL}(q(\mathbf{z}_t^{(i)} | \mathbf{x}_{1:t}, \mathbf{z}_{1:t-1}^{(i)}) \| p(\mathbf{z}_t^{(i)} | \mathbf{x}_{1:t-1}, \mathbf{z}_{1:t-1}^{(i)}))}_{\text{regularization loss}} \right) + \mathcal{L}_{\text{contrast}},$$
(13)

where the first and second terms correspond to $\mathcal{L}_{\text{P}}$ and $\text{D}_{\text{KL}}$ in the main text, respectively.

Since we assume a Poisson distribution for the inferred neural activity, $\mathcal{L}_{\text{P}}$ is the Poisson negative log-likelihood:

$$
\begin{aligned}
\mathcal{L}_{\text{P}}(\mathbf{x}_t, \mathbf{r}_t) &= -\log \frac{\mathbf{r}_t^{\mathbf{x}_t}}{\mathbf{x}_t!} e^{-\mathbf{r}_t} \\
&= -\mathbf{x}_t \log \mathbf{r}_t + \mathbf{r}_t + \log \mathbf{x}_t! \\
&\approx -\mathbf{x}_t \log \mathbf{r}_t + \mathbf{r}_t + \mathbf{x}_t \log \mathbf{x}_t - \mathbf{x}_t + \frac{1}{2} \log (2\pi \mathbf{x}_t).
\end{aligned}
$$
(14)

As for $\text{D}_{\text{KL}}$, under the assumption that both the prior and the approximate posterior are Gaussian, we have:

$$
\begin{aligned}
\text{D}_{\text{KL}}(\mathbf{z}_t^{(i)} \| \tilde{\mathbf{z}}_t^{(i)}) &= \int q(\mathbf{z}_t^{(i)} | \mathbf{x}_{1:t}, \mathbf{z}_{1:t-1}^{(i)}) \log \frac{q(\mathbf{z}_t^{(i)} | \mathbf{x}_{1:t}, \mathbf{z}_{1:t-1}^{(i)})}{p(\mathbf{z}_t^{(i)} | \mathbf{x}_{1:t-1}, \mathbf{z}_{1:t-1}^{(i)})} d\mathbf{z}_t^{(i)} \\
&= \int q(\mathbf{z}_t^{(i)} | \mathbf{x}_{1:t}, \mathbf{z}_{1:t-1}^{(i)}) \log \frac{\frac{1}{\sqrt{2\pi\boldsymbol{\sigma}_{z,t}^2}} \exp \left( -\frac{\left( \mathbf{z}_t^{(i)} - \boldsymbol{\mu}_{z,t} \right)^2}{2\boldsymbol{\sigma}_{z,t}^2} \right)}{\frac{1}{\sqrt{2\pi\tilde{\boldsymbol{\sigma}}_{z,t}^2}} \exp \left( -\frac{\left( \mathbf{z}_t^{(i)} - \tilde{\boldsymbol{\mu}}_{z,t} \right)^2}{2\tilde{\boldsymbol{\sigma}}_{z,t}^2} \right)} d\mathbf{z}_t^{(i)} \\
&= -\frac{\mathbb{E}_{q(\mathbf{z}_t^{(i)})} \left[ \left( \mathbf{z}_t^{(i)} - \boldsymbol{\mu}_{z,t} \right)^2 \right]}{2\boldsymbol{\sigma}_{z,t}^2} + \frac{\mathbb{E}_{q(\mathbf{z}_t^{(i)})} \left[ \left( \mathbf{z}_t^{(i)} - \tilde{\boldsymbol{\mu}}_{z,t} \right)^2 \right]}{2\tilde{\boldsymbol{\sigma}}_{z,t}^2} - \log \boldsymbol{\sigma}_{z,t} + \log \tilde{\boldsymbol{\sigma}}_{z,t} \\
&= -\frac{1}{2} + \frac{\boldsymbol{\sigma}_{z,t}^2 + \boldsymbol{\mu}_{z,t}^2 - 2\boldsymbol{\mu}_{z,t}\tilde{\boldsymbol{\mu}}_{z,t} + \tilde{\boldsymbol{\mu}}_{z,t}^2}{2\tilde{\boldsymbol{\sigma}}_{z,t}^2} - \log \boldsymbol{\sigma}_{z,t} + \log \tilde{\boldsymbol{\sigma}}_{z,t} \\
&= \frac{1}{2} \left( -1 + \frac{(\boldsymbol{\mu}_{z,t} - \tilde{\boldsymbol{\mu}}_{z,t})^2 + \boldsymbol{\sigma}_{z,t}^2}{\tilde{\boldsymbol{\sigma}}_{z,t}^2} - \log \boldsymbol{\sigma}_{z,t}^2 + \log \tilde{\boldsymbol{\sigma}}_{z,t}^2 \right).
\end{aligned}
$$
(15)

Finally, we apply NT-Xent loss as the contrastive loss:

$$\mathcal{L}_{\text{contrast}} = -\log \frac{\exp \left( \text{sim} \left( \mathbf{z}^{(e)}, \mathbf{z}_{\text{pos}}^{(e)} \right) / \tau \right)}{\exp \left( \text{sim} \left( \mathbf{z}^{(e)}, \mathbf{z}_{\text{pos}}^{(e)} \right) / \tau \right) + \sum \exp \left( \text{sim} \left( \mathbf{z}^{(e)}, \mathbf{z}_{\text{neg}}^{(e)} \right) / \tau \right)},$$
(16)

where $\text{sim}(*, *)$ is the cosine similarity and $\tau$ is the temperature coefficient.

## C  Generating Procedure of Synthetic Datasets

**Non-temporal dataset.**    First, we generate labels $u_i$ from four uniform distributions on $[\frac{2i \times \pi}{4}, \frac{(2i+1) \times \pi}{4}], i \in \{0, 1, 2, 3\}$, in preparation for building four clusters. Second, for each cluster, we sample 2-dimensional latent variables $\mathbf{z}$ from independent Gaussian distribution with $(5 \sin u_i, 5 \cos u_i)$ as mean and $(0.6 - 0.5| \cos u_i|, 0.5| \cos u_i|)$ as variance. Third, we feed sampled

latent variables into a RealNVP network [16] to form firing rates of 100-dimensional observations and generate the synthetic neural activity from the Poisson distribution. Each cluster contains 4,000 samples. This dataset, being label-dependent, is used to evaluate the models' ability to construct discriminative latent variables.

**Temporal dataset.** We generate three dynamic latent variables from the Lorenz system, consisting of a set of nonlinear equations. The firing rates of 30 simulated neurons are then computed by randomly weighted linear readouts from the Lorenz latent variables. Synthetic neural activity is also generated from the Poisson distribution. The hyperparameters of the Lorenz system follow a previous work [54]. We run the Lorenz system for 1s (1ms for a time point) from five randomly initialized conditions. Each condition contains 20 trials. This dataset is for assessing the models' ability to represent temporal information.

## D Characters of Mouse Visual Neural Dataset

The full names and abbreviations of all cortical regions are listed in Table 3 and the number of neurons for all chosen mice is also presented.

Table 3: Characters of the neural dataset.

| Cortical Region | Abbreviation | Mouse 1 | Mouse 2 | Mouse 3 | Mouse 4 | Mouse 5 |
|---|---|---|---|---|---|---|
| primary visual cortex | VISp | 75 | 51 | 93 | 63 | 52 |
| lateromedial area | VISl | 39 | 30 | 56 | 38 | 20 |
| rostrolateral area | VISrl | 49 | 24 | 58 | 44 | 41 |
| anterolateral area | VISal | 42 | 51 | 43 | 71 | 46 |
| posteromedial area | VISpm | 62 | 90 | 17 | 19 | 64 |
| anteromedial area | VISam | 94 | 72 | 49 | 60 | 64 |

## E Number of Trainable Parameters of All Models

The number of model parameters is roughly proportional to the number of input neurons. We present the number of parameters for the Mouse 1 dataset in Table 4.

Table 4: The number of model parameters for the Mouse 1 dataset.

| | LFADS | pi-VAE | Swap-VAE | CEBRA | TE-ViDS-small | TE-ViDS |
|---|---|---|---|---|---|---|
| Number of parameters | 0.45M | 0.49M | 0.38M | 0.71M | 0.29M | 0.68M |

## F Training Setup

We list the information of training datasets and the hyperparameters of the model training in Table 5. For each model, we perform the grid search for the learning rate ($1.0 \times 10^{-5}$ to $1.0 \times 10^{-3}$) and the weights of each loss term (0.01 to 10) to achieve optimal performance on the validation set.

Table 5: The information of each dataset and the hyperparameters of the training.

| Dataset | Training Size | Validation Size | Test Size | Latent Dimension | Learning Rate | Optimizer |
|---|---|---|---|---|---|---|
| non-temporal synthetic dataset | 12800 | — | 3200 | 32 | 0.0005 | Adam |
| temporal synthetic dataset | 80000 | — | 20000 | 8 | 0.001 | Adam |
| visual neural dataset under natural scenes | 118000 | 14750 | 14750 | 128 | 0.0001 | Adam |
| visual neural dataset under natural movie | 28800 | 3600 | 3600 | 128 | 0.0001 | Adam |

Furthermore, we present the preprocessing implementation of training samples for all models.

**Visual neural dataset.** For the neural activity under natural scenes, each sample input to TE-ViDS is sequential data from 5 time points and the offset of positive samples from target samples is within $\pm 3$

time points. For the neural activity under the natural movie, each sample consists of 4 time points and the maximum absolute offset is 2. As for alternative models, pi-VAE and Swap-VAE take the neural activity of an independent time point as an input sample. LFADS processes samples with the same length as our model. For CEBRA, following the original approach [48], we take the surrounding points centered on the target point to form a sequence (5 time points for natural scenes and 4 for the natural movie).

**Non-temporal dataset.** Since this dataset does not involve temporal relationships, each sample can be conceptualized as a sequence comprising a single time step. In the context of contrastive learning for such data, the approach of defining positive pairs through temporal shifting is inapplicable. Instead, we adopt a strategy provided in CEBRA, selecting positive samples based on the distance of their respective labels, thereby implementing a form of supervised contrastive learning.

**Temporal dataset.** TE-ViDS and LFADS use 50ms of data as input, while the other models take data at one time point. The offset of positive samples is set to 5ms for our model.

All models for all datasets are trained on 1 GPU (NVIDIA A100).

# G  Additional Results of Alternatives on Synthetic Non-Temporal Dataset

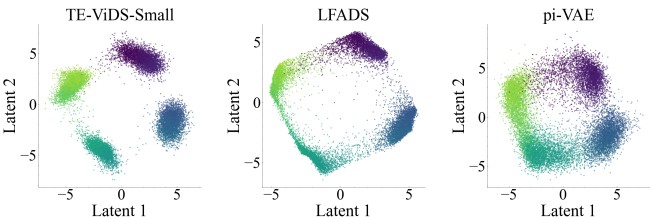

Figure 7: The inferred latent variables of alternatives on the non-temporal dataset.

# H  Additional Experiment on the Visual Neural Datatsets

We train models with neural activity under approximately 80% natural scene/movie stimuli and evaluate them using "held-out" stimuli, i.e., completely unseen scenes/movies. As Tables 6 and 7 show, TE-ViDS consistently achieves the best performance on such challenging tasks.

Table 6: The decoding scores (%) for "held-out" natural scenes.

| Models | Mouse 1 | Mouse 2 | Mouse 3 | Mouse 4 | Mouse 5 |
|--------|---------|---------|---------|---------|---------|
| LFADS | 48.96±1.13 | 34.78±1.83 | 39.22±2.47 | 38.17±1.60 | 15.74±1.14 |
| pi-VAE | 23.57±3.17 | 37.04±2.42 | 41.91±1.90 | 20.70±2.05 | 10.35±1.10 |
| Swap-VAE | 55.48±2.34 | 17.65±1.94 | 36.61±3.21 | 44.09±2.39 | 14.17±1.81 |
| CEBRA | 5.83±0.58 | 8.87±0.80 | 13.04±1.40 | 9.39±0.77 | 4.96±0.69 |
| **TE-ViDS-small** | 62.61±2.50 | **41.30±1.72** | **47.57±3.06** | 50.52±2.77 | 23.30±1.20 |
| **TE-ViDS** | **65.91±1.31** | 38.70±1.71 | 46.52±2.09 | **53.57±1.88** | **24.52±1.17** |

Table 7: The decoding scores (%, in 1s window) for "held-out" natural movie frames.

| Models | Mouse 1 | Mouse 2 | Mouse 3 | Mouse 4 | Mouse 5 |
|--------|---------|---------|---------|---------|---------|
| LFADS | 38.89±0.98 | 51.78±1.28 | 49.06±2.04 | 46.28±1.89 | 39.22±1.42 |
| pi-VAE | 38.89±1.34 | 65.39±0.95 | 66.56±1.31 | 59.11±1.34 | 46.17±1.13 |
| Swap-VAE | 36.22±1.83 | 59.72±1.35 | 58.33±1.62 | 57.72±1.77 | 44.50±1.19 |
| CEBRA | 37.44±1.04 | 54.39±1.41 | 52.89±0.84 | 48.89±0.86 | 37.89±1.60 |
| **TE-ViDS-small** | 39.28±2.15 | **68.06±1.09** | **72.50±0.81** | 65.50±0.78 | **47.83±1.47** |
| **TE-ViDS** | **42.00±0.52** | 66.28±1.07 | 69.06±0.84 | **66.28±1.43** | 47.00±1.80 |

# I   Additional Results of Latent Trajectories on the Visual Neural Datatsets

For the mouse neural dataset under natural scene stimuli, we visualize the latent representations by embedding them in two dimensions using tSNE (Figure 8). We focus on Mouse 1, for which most of the models achieve the highest scores. We select ten scenes that elicit the strongest average responses for visualization. Specifically, we reduce the dimensions of latent representations at a single time point and take the average across all trials, to show the latent trajectories over time for each scene. The result of TE-ViDS exhibits a clear temporal structure for different natural scenes. For LFADS, its ability to encode temporal features of sequential neural activity results in clear temporal structures, but the latent representations of different classes are largely intermingled. For pi-VAE, Swap-VAE, and CEBRA, their latent trajectories show varying degrees of entanglement over time. These results suggest that our model effectively distinguishes between category information and captures temporal information from neural dynamics well.

For the mouse neural dataset under natural movie stimuli, in addition to Figures 4C-E, we visualize the results of all models and all three parts of the movie for Mouse 2 (Figure 9).

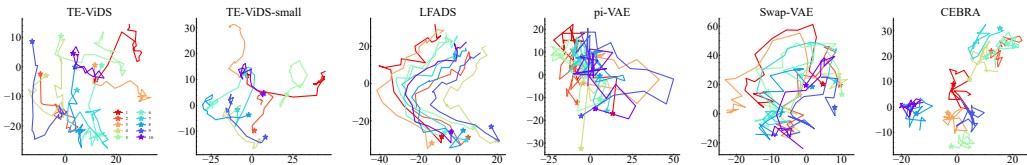

Figure 8: Visualization results of latent trajectories on the mouse visual neural dataset under natural scene stimuli (Mouse 1).

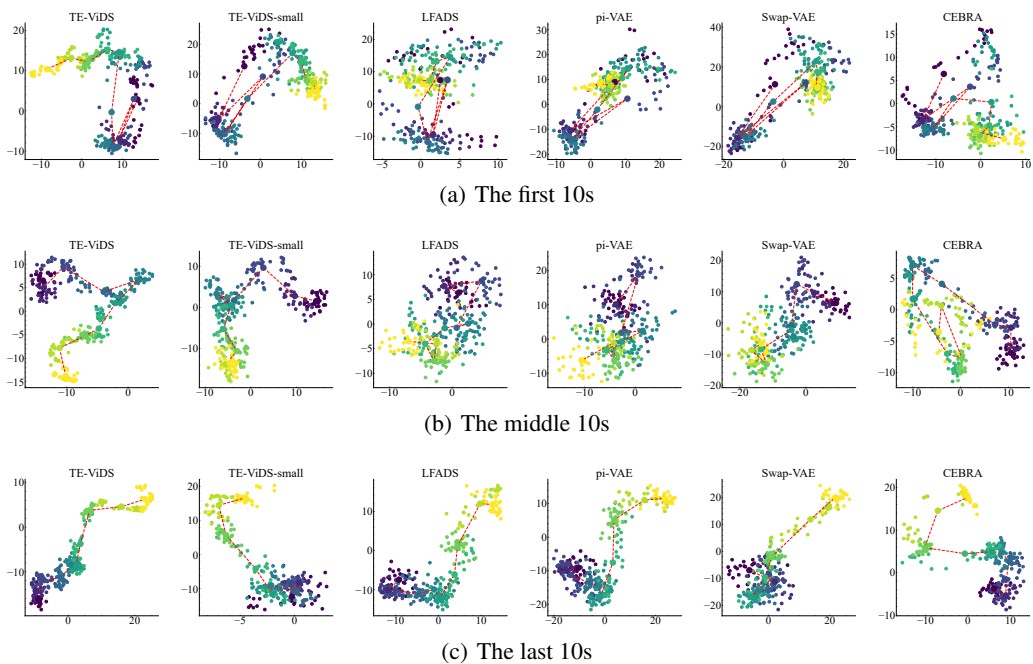

Figure 9: Visualization results of latent trajectories on the mouse visual neural dataset under natural movie stimuli (Mouse 2).

## J   Ablation Studies

We perform ablation studies for several aspects of TE-ViDS's components and neural activity input dimensions to explore their impact on performance.

**The loss function and the recurrent module of TE-ViDS.** To show the effectiveness of the components of our model, we conduct some ablation studies on the loss function and the recurrent

Table 8: The decoding scores (%) of ablation studies for the loss function, the recurrent module and the split design of TE-ViDS on the mouse visual neural dataset under natural scene stimuli.

| Models | Mouse 1 | Mouse 2 | Mouse 3 | Mouse 4 | Mouse 5 |
|---|---|---|---|---|---|
| **TE-ViDS** | **50.86±0.81** | **27.24±0.47** | **29.90±0.43** | **38.05±0.53** | **9.44±0.20** |
| w/o negative samples | 21.85±1.86 | 9.71±0.72 | 12.32±1.22 | 20.03±1.26 | 6.20±0.43 |
| w/o contrastive loss | 30.17±1.74 | 8.24±0.33 | 13.76±0.87 | 23.42±0.79 | 7.19±0.28 |
| w/o swap operation | 28.80±0.86 | 14.27±0.42 | 18.92±0.93 | 15.97±0.85 | 4.68±0.24 |
| w/o contrastive loss and swap operation | 6.88±0.64 | 3.92±0.22 | 4.17±0.72 | 6.58±0.32 | 2.56±0.12 |
| with temporal independent prior | 28.97±1.16 | 9.05±0.64 | 15.10±0.98 | 15.54±0.69 | 3.97±0.32 |
| GRU→Vanilla RNN | 47.64±0.78 | 25.61±0.68 | 28.58±0.66 | 34.15±0.43 | 7.17±0.30 |
| GRU→LSTM | 49.97±0.50 | 26.92±0.59 | 29.31±0.23 | 36.73±0.41 | 9.24±0.27 |
| Non-recurrent | 33.32±0.99 | 21.78±0.37 | 22.63±0.47 | 27.27±0.64 | 7.69±0.40 |
| External-Only | 45.41±1.16 | 21.49±0.46 | 23.83±0.73 | 27.39±0.67 | 6.80±0.23 |
| Internal-Only | 2.10±0.21 | 1.78±0.19 | 1.44±0.16 | 2.49±0.22 | 1.46±0.12 |

Table 9: The decoding scores (%, in 1s window) of ablation studies for the loss function, the recurrent module and the split design of TE-ViDS on the mouse visual neural dataset under natural movie stimuli.

| Models | Mouse 1 | Mouse 2 | Mouse 3 | Mouse 4 | Mouse 5 |
|---|---|---|---|---|---|
| **TE-ViDS** | **13.88±0.19** | **65.38±0.36** | 59.88±0.72 | 54.33±0.54 | 30.18±0.40 |
| w/o negative samples | 11.27±0.36 | 49.59±1.18 | 45.67±0.60 | 44.17±0.42 | 19.93±0.35 |
| w/o contrastive loss | 11.24±0.23 | 47.98±0.90 | 44.09±0.67 | 44.02±0.47 | 18.24±0.41 |
| w/o swap operation | 10.22±0.31 | 49.39±0.57 | 45.30±0.41 | 43.12±0.57 | 21.83±0.39 |
| w/o contrastive loss and swap operation | 9.16±0.37 | 24.84±1.10 | 22.33±0.81 | 26.49±0.66 | 12.02±0.49 |
| with temporal independent prior | 12.09±0.16 | 57.74±0.62 | 53.87±0.49 | 46.90±0.48 | 22.92±0.43 |
| GRU→Vanilla RNN | 13.08±0.31 | 63.19±0.55 | 59.13±0.42 | 53.83±0.41 | 29.62±0.40 |
| GRU→LSTM | 12.77±0.25 | 64.69±0.53 | **60.00±0.60** | **54.37±0.41** | 28.50±0.61 |
| Non-recurrent | 11.14±0.30 | 53.26±0.48 | 48.31±0.47 | 43.81±0.40 | 22.98±0.44 |
| External-Only | 12.16±0.24 | 63.33±0.29 | 58.64±0.37 | 52.11±0.39 | **30.24±0.54** |
| Internal-Only | 7.57±0.24 | 11.73±0.43 | 12.86±0.35 | 16.10±0.33 | 9.89±0.22 |

module (Tables 8 and 9). In terms of contrastive learning, we first exclude negative samples from the computation of the contrastive loss and use only the cosine distance between the *external* latent variables of positive sample pairs as the loss function. In other words, we only bring the positive pairs closer. This results in a decrease in performance, suggesting that negative samples are useful. Next, we remove either the contrastive loss or the swap operation and observe a similar impact for both. Finally, we remove both, meaning that there are no more objectives related to contrastive learning. The significant decrease in performance suggests that contrastive learning plays a crucial role in our models. In terms of the regular loss of the *internal* latent variables, we originally assumed that the prior distribution is time-dependent. When we assume that it is an independent standard normal distribution at each time step, the model's performance degrades, demonstrating that time-dependent assumptions about the prior are also important. In terms of the recurrent module, the results suggest that GRU is a better choice when considering the trade-off between performance and computational efficiency. Besides, we evaluate a non-recurrent version of our model by setting the time steps of GRU to 1, which demonstrates that the recurrent module plays a critical role.

**The split design of latent variables.** We build two models based on TE-ViDS, one with *external* variables only (External-Only) and the other with *internal* variables only (Internal-Only). Both models perform worse than TE-ViDS, confirming the split design's importance and effectiveness.

**The dimension of latent variables and the number of input neurons.** We perform ablation studies on the number of latent variables and input neurons. As shown in Figures 10 and 11, first, the performance saturates gradually as the dimension of the latent variables increases, which suggests that it is sufficient to choose a dimension in a reasonable range (not much fewer than input neurons). Second, performance decreases as the number of sampled neurons decreases, suggesting that for each mouse, all recorded neurons contribute to the representation of visual stimuli.

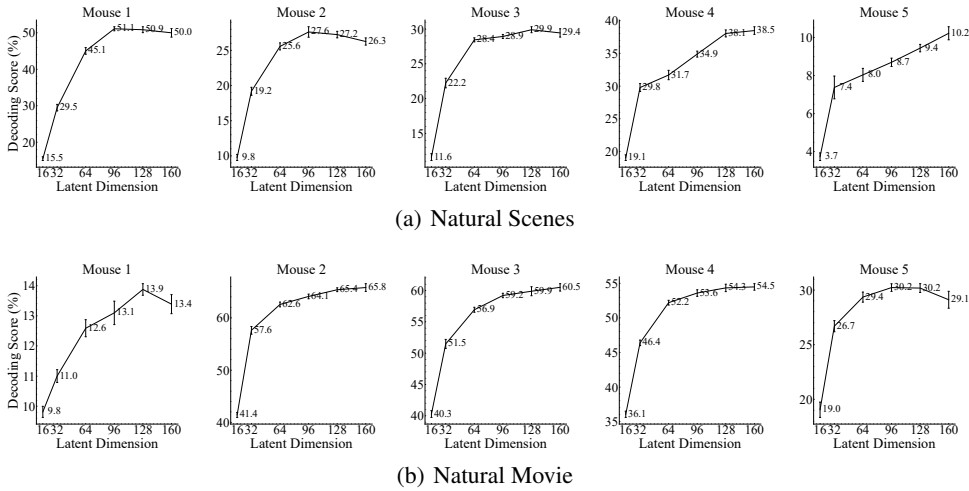

(a) Natural Scenes

(b) Natural Movie

Figure 10: The results of ablation studies on the dimension of latent variables.

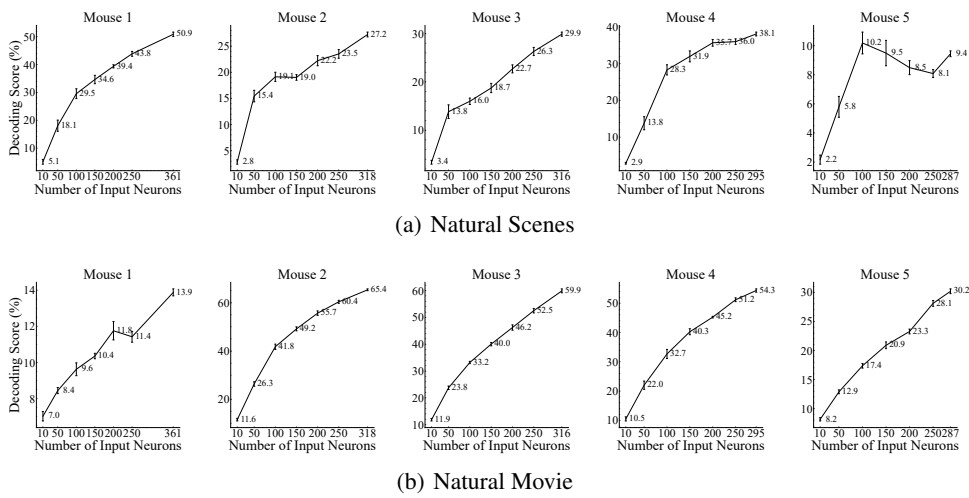

(a) Natural Scenes

(b) Natural Movie

Figure 11: The results of ablation studies on the number of input neurons.

# K  Additional Experiment on the Mouse Visual Neural Dataset from CEBRA

The neural dataset used in CEBRA is also preprocessed from the Allen Brain Observatory Visual Coding dataset. The dataset is generalized by sampling different numbers of neurons from the same visual cortical region of all mice, without taking into account the variability between subjects. We evaluate our model on this dataset. The results (Figure 12) show that our model performs better on the primary and mid-level regions, while CEBRA performs better on the high-level regions. Moreover, in the case of a sample with 40 time steps (CEBRA reported their highest performance in this setting), our model can achieve a performance of more than 95% with fewer trainable parameters (TE-ViDS: 0.44M; CEBRA: 1.09M).

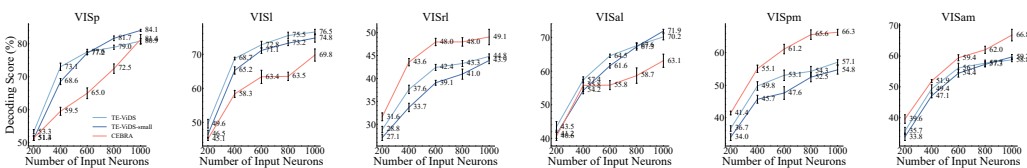

Figure 12: The decoding scores (%, in 1s window) on the mouse neural dataset used in CEBRA.

