# OpenReview forum: "Time-Evolving Dynamical System for Learning Latent Representations of Mouse Visual Neural Activity"
_NeurIPS.cc/2025/Conference — NeurIPS 2025 poster_

### Official Review · Reviewer_RfuT · 2025-06-27

**Clarity:** 3
**Significance:** 2
**Originality:** 2
**Rating:** 2
**Confidence:** 5

**Summary:**

Time-Evolving Dynamical System for Learning Latent Representations of Mouse Visual Neural Activity

This paper presents a method for time-varying representations in a neural network for visual data decoding in a latent space. The method essentially relies on two RNNs - one of which is regularized towards external behavioral variables and another that is regularized with a contrastive loss to be "smooth". The method essentially is a mild variation of existing work, some of which is cited but many others that are omitted. I think really understanding what this method has to offer would require much more rigorous comparison to the many editing dynamical system LVM models.


Major comments:

"most work on LVMs has not explicitly considered neural temporal relationships"
Off the top of my head here are the models I can think of:
 - LDS: https://pubmed.ncbi.nlm.nih.gov/22663075/
 - POGLM: https://arxiv.org/pdf/2402.01263
 - PLINE: https://proceedings.mlr.press/v139/kim21h.html
 - DPAD: https://www.nature.com/articles/s41593-024-01731-2
 - rsLDS: https://arxiv.org/abs/1610.08466
 - CREIMBO: https://openreview.net/forum?id=28abpUEICJ
 - dLDS: https://jmlr.org/papers/v25/23-0777.html
 - PSID: https://pubmed.ncbi.nlm.nih.gov/33169030/
 - DFINE: https://pubmed.ncbi.nlm.nih.gov/38082181/
 - GP-VAE: https://arxiv.org/pdf/2310.03111
 - RP-GSSM: https://arxiv.org/abs/2505.23569
 - POYO: https://openreview.net/forum?id=sw2Y0sirtM


Many of the above are much more reasonable to compare to - especially the both regularized AND sequential (like PSID or DFINE) than the comparisons provided in the paper.

The disentangling is a major part of the paper and is similar to PSID (sequential - cited above) and the now many nonlinear CCA style methods (supervised - examples listed below). These comparisons may be more appropriate than CEBRA.
 - https://proceedings.mlr.press/v28/andrew13.html
 - https://openreview.net/forum?id=5FUq05QRc5b
 - https://arxiv.org/abs/2408.12091
 - https://arxiv.org/abs/2312.13455

**Questions:**

I'm not really a fan of CEBRA but even I find the decoding scores in Figure 3 suspiciously low (also for pi-VAE). Do the authors have an explanation for that? It seems like something significant went wrong, potentially a bug?

The separation of internal and external stimuli can be challenging and similar methods (DCCA, Lyu et al, SPLICE, etc listed above) rely on information theoretic approaches to quantify disentangling. Here the primary metric is the decoding of external variables. Is this sufficient? Would estimates of the information show that these numbers are low enough to consider the internal variables truly internal?

**Ethical Concerns:**

["NO or VERY MINOR ethics concerns only"]

**Final Justification:**

While I think that the authors did try hard to address the comments, there are a number of new baselines and I feel that the paper needs significant editing to incorporate these changes. Unfortunately, it's very hard to assess these new baselines given the NeurIPS format and I think that a new submission to the next conference would be much stronger.

**Limitations:**

Limitations are not discussed, and it seems at the very least the limitation of data requirements to fit this model should be discussed (especially when there are linear system alternatives that would take much less data to fit).

**Paper Formatting Concerns:**

Only that there is no explicit limitations section.

**Quality:**

2

**Strengths And Weaknesses:**

Strengths:
 - The model derivation seems correct
 - The results for the model in absolute terms seem interesting
Weaknesses
 - The comparisons are lacking in appropriate baselines
 - There are many missing references
 - The novelty of the model is fairly limited

---

> ### Author Rebuttal · Authors · 2025-07-31
>
> We thank the reviewer for the pointed and constructive comments. We will do our best to address the comments and answer the questions.
>
> **R1. References and baselines.**
>
> We sincerely appreciate the reviewer's valuable suggestions regarding additional Latent Variable Model (LVM) references, and we will gladly incorporate a discussion of these works into our revised manuscript. However, we respectfully contend that these suggested models may not be more appropriate for direct comparison with our method than the baselines we have already employed.
>
> First, our work proposes a novel visual dynamical system that leverages self-supervised contrastive learning to disentangle neural activity components related to visual stimuli from those influenced by internal brain states. A core characteristic of our approach is the absence of any supervised signals (e.g., external behavioral variables) in the model's training. Given these specific model characteristics, we selected our current baselines. LFADS is a classic and widely recognized dynamical system, providing a robust benchmark for temporal neural data. pi-VAE incorporates external supervised information (and supporting discrete variables) to construct interpretable latent variables, contrasting with our self-supervised objective. Swap-VAE utilizes contrastive learning to construct separated latent variables, aligning with our disentanglement goal, although it does not explicitly model temporal dynamics. CEBRA also employs contrastive learning for latent variable construction. Critically, the work on CEBRA has conducted extensive analyses of neural activity in the mouse visual cortex and our experimental setup for visual neural activity largely follows its established methodology. This direct alignment in an experimental context makes CEBRA a particularly pertinent baseline.
>
> Second, most of the models mentioned by the reviewer are either oriented towards other research topics or possess characteristics that render them less applicable to our specific context. For instance, models such as LDS and PLINE primarily focus on modeling the dynamics of synthetic data or motor behaviors in the motor cortex. Similarly, PSID, DFINE and DPAD are designed to separate behavior-relevant and behavior-irrelevant neural dynamics using supervised information, which differs from our self-supervised approach. MM-GPVAE is tailored for multimodal inputs, while CREIMBO and POYO focus on modeling population dynamics across brain regions or individuals. Finally, models like RP-GSSM and DCCA are primarily concerned with learning low-dimensional representations of images or videos, which fall outside the scope of our neural activity analysis.
>
> Nevertheless, as suggested by the reviewer, we incorporate PSID, DFINE, and DCCA into our experiments on visual neural activity for comparison. Notably, the supervised versions of PSID and DFINE leverage continuous, high-temporal-resolution behavioral variables as supervised information to separate behavior-relevant and behavior-irrelevant neural dynamics. This approach presents significant challenges for application to the disentanglement of visual-related and visual-unrelated neural activity in our context. Specifically, biological experiments often utilize static visual stimuli, or dynamic stimuli with temporal resolutions considerably lower than that of neural activity. Furthermore, natural visual stimuli are high-dimensional and complex signals. The various methods of constructing visual information for supervision introduce strong human-induced priors that could influence the nature of the constructed latent variables. For example, using pixel values, low-order visual features, or high-order semantic information for supervision corresponds to visual information encoded by different regions of the visual pathway. Therefore, we adopt the unsupervised versions of PSID and DFINE. As presented in Table r1, our model consistently demonstrates superior performance compared to these models.
>
> Table r1: The results (\%) for decoding natural movie frames.
>
> ||Mouse 1|Mouse 2|Mouse 3|Mouse 4|Mouse 5|
> |-|:-:|:-:|:-:|:-:|:-:|
> |PSID|8.11|38.33|39.56|29.67|14.11|
> |DFINE|10.36±0.56|55.53±0.71|46.56±0.38|45.49±0.68|22.96±0.63|
> |DCCA|12.76±0.54|60.13±0.64|58.47±0.87|43.31±1.42|26.82±1.11|
> |TE-ViDS|13.88±0.19|65.38±0.36|59.88±0.72|54.33±0.54|30.18±0.40|
>
> **R2. The low performance of CEBRA in Table 1.**
>
> We think that CEBRA demonstrates stronger performance in distinguishing between coarse-grained categories but less capability for finer-grained distinctions. First, this is suggested by Figure 8 in the Appendix, where, despite CEBRA's capacity to differentiate some categories through contrastive learning, considerable overlap is observed in the neural latent representations corresponding to other categories. This limitation is further evinced in the movie decoding experiment (Figure 4B), where CEBRA's performance declines more significantly as the constraint window narrows (i.e., an increased number of movie frame categories).
>
> To corroborate this view, we further perform an experiment on static scene decoding. We train the models on neural data of the five natural scenes that elicit the strongest neural responses and evaluate their performance on the five-classification task. As shown in Table r2, although CEBRA's performance remains inferior to our model, it achieves a relatively good level, supporting our hypothesis regarding its better performance on coarser distinctions.
>
> Furthermore, we note that the lower performance of pi-VAE is confined to Mouse 1, with its performance on other mice being at an intermediate level, thus ruling out the possibility of a bug.
>
> Table r2: The results (\%) for decoding five natural scenes.
>
> ||Mouse 1|Mouse 2|Mouse 3|Mouse 4|Mouse 5|
> |-|:-:|:-:|:-:|:-:|:-:|
> |LFADS|79.6±2.8|78.4±4.3|68.0±4.3|71.2±1.7|61.6±3.6|
> |pi-VAE|97.2±1.0|81.2±1.8|80.8±1.4|85.6±2.1|69.2±2.0|
> |Swap-VAE|98.0±0.8|69.6±2.3|77.2±1.6|76.4±2.3|84.8±2.2|
> |CEBRA|96.4±1.3|76.8±1.9|80.4±1.2|72.8±1.9|56.8±2.7|
> |TE-ViDS|100.0±0.0|95.6±1.2|94.8±1.1|94.8±0.6|90.4±1.0|
>
> **R3. The evaluation for internal latent variables.**
>
> While our current findings indicate a low correlation between internal latent variables and visual stimuli, we acknowledge the absence of quantitative metrics in this work to definitively verify if these variables reflect an animal's internal state. To address this limitation, we will utilize more comprehensive datasets (e.g., incorporating behavioral and physiological recordings) for robust evaluation in the future.
>
> **R4. The data requirements.**
>
> This represents a trade-off: our model, being more complex than linear alternatives, incurs higher computational costs when applied to smaller datasets. However, for larger datasets, where linear models are prone to underfitting, our model demonstrates superior scalability and the capacity to capture more intricate relationships.

---

> > ### Comment · Reviewer_RfuT · 2025-08-03
> > **Still not sure about the exact placement in the literature**
> >
> > I appreciate the author's response.
> >
> > R1: I continue to disagree with the dismissal of other models. A model being applied to a different brain area does not discount it as a model with properties similar to the proposed method. This is a machine learning conference after all and what matters is more what model capabilities are being proposed and how they compare to other models' capabilities - not what data they've been applied to. I also found parts of the response a bit off For example "LDS and PLINE primarily focus on modeling the dynamics of synthetic data or motor behaviors in the motor cortex". I'm not sure what "modeling the dynamics of synthetic data" means or why LDS - one of the most widely used models across pretty much all domains that deal with dynamical data, is being pigeon-holed in this way. I appreciate the additional runs, however I do think that rather than comparing just to what people use, comparing to what has the most similar goals is the more appropriate comparison. Hence, e.g., why ablation studies on complex models are so informative.
> >
> > R2: I appreciate the clarification and the additional tests. This makes sense to me.
> >
> > R3: I do believe that quantitative metrics will be critical in convincing readers that the model is truly accomplishing the stated goals.
> >
> > R4: Is there any idea of where the trade-offs happen? What data level is needed for overfitting this nonlinear method to stop overfitting, e.g., as opposed to where the performance of the linear system to fall?

---

> > > ### Author Response · Authors · 2025-08-04
> > >
> > > Thank the reviewer for the feedback. We will provide more clarification on the reviewer's concerns.
> > >
> > > 1). Our baseline selection, as detailed in R1, **exactly prioritizes models with properties similar or related to our own**, rather than those applied to the same dataset (only CEBRA conducted experiments on the same dataset). We agree on the significant influence of LDS models; our observation that a majority of the experiments in the mentioned studies (e.g., rSLDS and dLDS) are conducted on synthetic data is not a denial of their capacity to extract neural dynamics from real neural data. Furthermore, we have provided the additional comparisons requested by the reviewer, which demonstrate our model's superiority. We are open to further discussion should any concerns regarding these comparisons persist. Finally, as clarified above, we do not focus on "models that people use", but rather on models that share similar characteristics or objectives with our model. Regarding the ablation studies, we have provided comprehensive results in Appendix J.
> > >
> > > 2). Validation of the trade-off requires evaluation across more datasets of varying sizes. Here, we present an initial observation. In our current work, we utilize two datasets, where each mouse has approximately 300 neurons. The first, comprised of 118 natural images displayed for 250 milliseconds and repeated 50 times, results in a total of 1,475 seconds of neural activity. The second, a 30-second natural movie displayed 10 times, yields a total of 300 seconds of neural activity. In both cases, we do not observe overfitting in our model. However, when the model is trained on a subset of five natural images, totaling 62.5 seconds of neural activity (as presented in R2), it exhibits overfitting. We mitigate this by implementing early stopping based on validation set loss. Additionally, we find that reducing the model's depth and width (TE-ViDS-small), while maintaining its core properties, is effective in counteracting overfitting. This demonstrates that our model's scalability is robust and adaptable to different data sizes.

---

### Official Review · Reviewer_iXM1 · 2025-06-27

**Clarity:** 2
**Significance:** 3
**Originality:** 3
**Rating:** 4
**Confidence:** 4

**Summary:**

This work presents a latent variable model for time-varying neural activity in mice, using an encoder-decoder architecture with a temporally evolving latent space split into two components: “external” and “internal.” The external component is a deterministic function of neural activity, aimed at capturing stimulus-dependent information, while the internal component is stochastic and intended to capture neural variability. The model is trained using a combination of a variational lower bound and a contrastive loss. On synthetic datasets, it shows improved latent and stimulus reconstruction. Applied to the large-scale Allen Brain Observatory Visual Coding dataset, the model outperforms baselines on natural scene and movie decoding, where decoding is framed as a scene/frame classification task.

**Questions:**

1. Why use the contrastive loss and only on z^e? It’s not clear to me what the motivation there is precisely.

2. I understand that the combination of the reconstruction term and KL divergence forms the standard ELBO. However, it’s unclear how the full objective, which includes the additional contrastive loss, is still considered an ELBO. In Appendix B, you derive the ELBO and then add a negative contrastive loss term, but it’s not clear whether the ELBO inequality still holds under this modified objective. As I understand it, the contrastive term is not necessarily independent of the ELBO (z^{e} appears in the reconstruction loss)—so minimizing the contrastive loss could potentially conflict with maximizing the ELBO. It would be helpful if you could clarify how these objectives interact and whether the resulting objective still maintains a valid ELBO interpretation.

3. There seems to be substantial variation in decoding performance across the different mice (esp mouse 5 vs rest in Figure 3 and mouse 1 vs rest in Figure 4). Do you have any hypotheses as to why this might be the case?

4. What are the main limitations of your model and approach? What future directions are you considering? Are there specific neuroscientific questions you aim to address with your model?

Please also see questions under "weaknesses".

**Ethical Concerns:**

["NO or VERY MINOR ethics concerns only"]

**Final Justification:**

The authors have addressed concerns about the work’s positioning in the literature, provided explanations for model and loss design, and agreed to rewrite parts of the model description for clarity. They have also extended their evaluation by including log-likelihood scores (ELBO) showing superior performance over comparable models in the literature.

However, three concerns remain. First, the quality of the learned internal latent variable is unclear—while it shows a weak correlation with stimulus, the work could be strengthened by also measuring correlations with observed behaviors and demonstrating a stronger trend. Second, the choice of synthetic datasets is relatively simple; more expressive datasets would better test the model’s ability to uncover latent factors alongside stimulus information. Third, although the authors argue that adding the contrastive term to the loss does not violate the ELBO, this still needs to be formally proven or reproduced from prior work.

Overall, I find the work valuable and would increase my score commensurate with how effectively these points are addressed.

**Limitations:**

Limitations of the model and approach are not discussed. Please see my question 4.

**Quality:**

3

**Strengths And Weaknesses:**

### Overall assessment

Edit: note that this assessment was written prior to the rebuttal phase. See below for final justification.

This paper presents a promising direction for latent variable modeling of neural activity but is currently significantly limited by unclear motivation, under-explained methods, and insufficiently rigorous evaluation.

**To help me raise my scores**, I encourage the authors (where feasible) to: **(1)** clarify the model architecture and its novelty; **(2)** justify the use of the contrastive loss and its interaction with the ELBO; **(3)** substantiate the choice of simulated data and corresponding evaluation; **(4)** strengthen the evaluation with more rigorous metrics (e.g., log-likelihood/ELBO on spikes), justify the decoding setup; **(5)** clearly articulate the contribution relative to prior work, and importantly **(6)** reflect and discuss the paper's limitations.

Details below.

## Strengths

- This work tackles an important and timely problem in computational neuroscience and machine learning—analyzing time-varying neural activity in response to natural and complex visual stimuli—by developing a latent variable generative model of the neural activity.
- The model explicitly separates stimulus-related and internally driven components in a temporally evolving latent space leading to more structured and interpretable representations of neural activity.
- The paper presents results on the publicly available, large-scale Allen Brain Observatory Visual Coding dataset consisting of neural activity across six different cortical regions and five different mice, recorded using modern Neuropixel probes. It also includes comparisons with several existing latent variable models, making it overall a suitable benchmark for evaluating approaches to modeling mouse visual neural activity.

## Weaknesses

**Summary**: The core contributions, motivation, and methodological choices are often unclear or underexplained. Key concerns include vague problem framing, limited detail on synthetic datasets, coarse decoding evaluation with scalability issues, and missing assessments of spike reconstruction and internal latent utility. The writing would also benefit from revision for clarity, grammar and style.

1. **Lack of clarity in problem motivation, methodological contribution, and model positioning:** In the initial sections—abstract, introduction, and related work—I found it difficult to identify the core methodological contribution, the specific scientific question being addressed, and the gap in prior work that the model aims to fill. While these sections hint at high-level goals, such as using latent variable models (LVMs) to capture correlations between neural activity and visual stimuli or behavior, they do not clearly present the motivation for applying LVMs in this context, provide a concise description of the proposed approach, or explain how it meaningfully differs from existing models. The rationale behind key design choices, particularly in relation to modeling neural activity to visual stimuli, is also unclear. A more focused and structured presentation of these elements before diving into the technical details would greatly enhance the clarity and accessibility of the paper.
2. **Lack of clarity and motivation in model architecture and proposed methodology:** In Section 3, the method is not clearly motivated or presented in terms of the model architecture and the specific relationships between variables. While Figure 1 offers a helpful high-level overview, it omits key variables such as $h_t$, $x_t$, $z_t^e$, $z_t^i$, and $r_t$, which are central to the model. Although the text outlines how these variables interact, the rationale behind their roles and the choices made is not clearly explained, making it difficult to develop an intuitive understanding. The appendix provides a graphical representation of the model, but it lacks a descriptive summary of the relationships among these variables and the motivation for these specific design choices, especially given that the model being proposed is quite complex. The same goes for the loss; why exactly include the contrastive loss? For clarity and completeness, I would recommend the authors include a concise and self-contained explanation of the model and its components directly in the main text, even if some fundamental choices are based on previous work. Important, why are these choices central to the context of neural activity in response to visual stimuli (if they are intended as such) and how are they exactly different from the baselines considered? This would also substantially help understand these specific baselines were chosen, which at the moment feels not well-justified.
3. **Synthetic data lacks sufficient detail**: The descriptions of the synthetic datasets lack essential detail. The current explanation—“[Line 166] Synthetic datasets roughly point to two properties of visual neural activity…”—is vague and uninformative, and does not clarify what the synthetic data actually looks like, how it’s constructed, or what aspects of visual neural activity it is intended to model.
4. **Unclear whether synthetic results evaluate the model’s core contributions:** When it comes to the results, while they show that their model is superior in reconstructing the true latents and the observations, it is not clear whether it tests their model’s novelty — be it the split internal-external latent variables or their loss functions — or its relevance to neural activity to visual stimuli. It’s difficult to judge how good of an evaluation this represents for their proposed model.
5. **Decoding task treated as coarse grained classification with limited scalability:** In their main evaluation on real data, the authors frame the task as decoding natural scenes and movie frames, but in practice, they treat it as a classification problem: predicting which labeled image (out of 118 natural scenes) or movie frame (out of 900) was shown to the animal. For movie frames, predictions are considered correct if they fall within +/-1 second—about 30 frames if you consider 30 fps—making this a coarse-grained temporal task rather than frame-level decoding. This approach also doesn’t scale well to longer videos or larger scene sets that are typical, where the label space becomes unwieldy. Additionally, choices like searching for K (in their KNN) in odd numbers from 1 to 20 are ad hoc and not well justified.
6. **Missing evaluation of spike reconstruction and internal latent utility:** While it is encouraging to see the model outperform all baselines on scene and frame decoding, with the external latent clearly showing higher stimulus decodability, the paper does not evaluate the model’s ability to reconstruct spike counts. Given that the core task is to model the distribution of spike counts—albeit through latent variables—it would be more rigorous to report the log-likelihood (or ELBO) on held-out spike data and compare it to other models that support such evaluation. Additionally, the paper lacks a benchmarking task to assess whether the “internal” latent component captures meaningful structure. Showing that it does not aid stimulus decoding seems to me insufficient to demonstrate its utility or interpretability.
7. **Scope for improvement in written clarity and style:** The paper would benefit from careful editing to better align with the tone and conventions of scientific writing. Many sentences feel unwieldy and imprecise:

(i) “[L38–39]: *Given that revealing the correlation between neural activity and visual stimuli is a challenging research highlight in the neuroscience community, it is desperately necessary to develop a strong model for extracting vision-related latent representations.*”

(ii) “[L48]: *The external latent representations aim to magnify stimulus-relevant components of visual neural activity, while the internal latent representations point to dynamical internal states that have an impact on the animal’s perceptive capability.*”

(iii) “[L117] *The high-quality latent representations are compressed information on the original neural activity, so they are needed to reconstruct the input neural activity well.*”

In above examples, the phrasing could be made more concise, neutral, and aligned with typical scientific expression.

---

> ### Author Rebuttal · Authors · 2025-07-31
>
> We thank the reviewer for the notable and perceptive comments. We will do our best to address the reviewer's concerns and answer the questions in the following.
>
> **R1. Problem motivation, the core methodological contribution and the gap in prior work.**
>
> **Problem motivation and scientific objective**: LVMs have proven highly influential in studies for the motor cortex, primarily by extracting low-dimensional neural embeddings that encode motor features and capture motor-related neural dynamics. However, LVMs have limited application in the analysis of visual neural activity. In our work, the scientific objective is to capture low-dimensional, stimulus-relevant embeddings from visual neural activity, particularly in the context of naturalistic visual stimuli. More specifically, we aim to leverage LVMs to disentangle neural components related to visual stimuli from those influenced by internal states. Through this disentanglement, we seek to further explain the observed variability in visual neural activity across subjects and across cortical regions.
>
> **The core methodological contribution**: We have provided a concise approach description in lines 45-50. Here, we make a more detailed clarification. First, we introduce temporal structures to explicitly establish temporal relationships for latent variables in a causal order. This design allows the latent variables to evolve over time, capturing the temporal dependencies inherent in neural activity. Second, we implement a split structure and design distinct loss functions to build two specialized sets of latent variables: external latent representations are trained to capture stimulus-relevant components of visual neural activity and internal latent representations are developed to reflect the animal's dynamic internal states.
>
> **The gap in prior work**: Existing studies via LVMs on visual neural activity primarily focus on simple visual stimuli and do not investigate extracting meaningful neural embeddings under naturalistic visual stimuli. More importantly, there was a lack of LVMs capable of splitting latent representations to explicitly differentiate between external stimulus-relevant and internal state-related components in visual neural activity in a self-supervised manner. For detailed differences in structure and design from previous models, please refer to R9 to Reviewer RtPx.
>
> **R2. Model architecture and methodology of model training.**
>
> **Model architecture**: Our model's core comprises external and internal latent variables, representing distinct visual neural activity components, alongside state variables for temporal processing. These variables interdepend through modules, fully described in the manuscript; here, we highlight design motivations.
> - Many component dependencies are adapted from the classic VRNN.
> - In the encoder, both external and internal latent variables are conditioned on input neural activity and their respective RNN state variables, crucial for causal temporal capture.
> - The internal latent variable's prior is conditioned solely on state variables, promoting internal state spontaneity.
> - The decoder reconstructs neural activity conditioned on both latent and internal state variables, with internal state variables enriching neural dynamics for more precise reconstruction.
> - Finally, external RNN state variables depend only on current input and previous state, while internal state variables also depend on latent variables, reflecting how external visual stimuli influence an animal's internal states.
>
> **Loss functions**: The objective of multiple training terms within our designed loss functions is to achieve disentanglement between external and internal latent variables, compelling them to reflect distinct components of visual neural activity.
> - As in lines 132–141, contrastive learning is applied to the external latent variables. It encourages clustering external latent variables corresponding to sample pairs with small temporal offsets (i.e., under similar visual stimuli), associating these variables with the visual input. Concurrently, a time-dependent prior distribution exhibiting spontaneity guides the internal latent variables via KL divergence. The differential application of these distinct loss functions to the two sets of latent variables is instrumental in promoting their disentanglement.
> - The swapping operation (detailed in R1 to Reviewer NU2a) further reinforces this disentanglement. It compels the model to strengthen the distinct roles of the external and internal latent variables, thereby enhancing their disentangled representation.
>
> For the reason for the baseline choices, please refer to R1 to Reviewer RfuT.
>
> **R3. The experiments of synthetic datasets.**
>
> **We have provided the detailed generation procedure for the two synthetic datasets in Appendix C**. Specifically, the non-temporal dataset is constructed using several label clusters to represent distinct visual stimulus categories, while the temporal dataset is designed to characterize the temporal relationships inherent in neural activity.
>
> These synthetic experiments are crucial for evaluating the fundamental ability of LVMs to construct accurate latent variables—a standard preliminary step in this field. The verification of our model's core contribution and its efficacy in extracting latent variables related to actual visual stimuli is subsequently accomplished through experiments with real visual neural activity datasets.
>
> **R4. About the decoding task.**
>
> For decoding tasks and coarse-grained (within +/-1 second) movie frame classification, **we largely follow CEBRA's established approach**, as it has conducted extensive analysis of mouse visual cortex activity. This alignment facilitates fair baseline comparisons. In addition, our model consistently outperforms others in finer-grained (the minimum is +/-0.2 second) frame classification, as shown in Figure 4B.
>
> Nevertheless, this approach has limitations with longer movies or larger scene datasets. Future work will explore more generalized metrics for evaluating latent variable-visual stimuli relationships, such as comparing the similarity between latent variables and low- or high-level visual features.
>
> For k-value selection, empirical values were derived considering dataset size, latent variable dimension, and category count. An odd $k$ value was chosen to reduce decision uncertainty.
>
> **R5. Evaluation of spike reconstruction and internal latent variables.**
>
> We evaluate spike reconstruction on the test set for models other than CEBRA. Our model consistently demonstrated superior performance, as indicated by lower Poisson negative log-likelihood values (Tables r1, r2).
>
> We acknowledge the current lack of quantitative metrics for internal latent variables. While preliminary results suggest low correlation with visual stimuli, future work will require more comprehensive datasets to rigorously validate whether these variables accurately reflect an animal's internal state.
>
> Table r1: The reconstruction results for the scene dataset.
>
> ||Mouse 1|Mouse 2|Mouse 3|Mouse 4|Mouse 5|
> |-|:-:|:-:|:-:|:-:|:-:|
> |LFADS|0.234±3.6e-5|0.222±4.5e-5|0.232±3.5e-5|0.208±5.8e-5|0.226±5.7e-5|
> |pi-VAE|0.252±8.7e-5|0.241±2.8e-5|0.248±2.4e-5|0.227±4.0e-5|0.245±7.1e-5|
> |Swap-VAE|0.202±2.1e-4|0.189±2.6e-4|0.196±2.5e-4|0.181±2.9e-3|0.196±7.3e-4|
> |TE-ViDS|0.195±7.6e-4|0.145±1.6e-4|0.184±4.9e-3|0.147±4.8e-3|0.174±1.3e-3|
>
> Table r2: The reconstruction results for the movie dataset.
>
> ||Mouse 1|Mouse 2|Mouse 3|Mouse 4|Mouse 5|
> |-|:-:|:-:|:-:|:-:|:-:|
> |LFADS|0.134±3.9e-4|0.196±5.1e-4|0.186±5.5e-4|0.192±5.3e-4|0.163±5.4e-4|
> |pi-VAE|0.137±1.7e-4|0.195±6.6e-5|0.185±7.7e-5|0.192±9.2e-5|0.163±7.8e-5|
> |Swap-VAE|0.100±2.7e-4|0.153±2.9e-4|0.144±2.4e-4|0.149±2.0e-4|0.119±2.3e-4|
> |TE-ViDS|0.086±4.0e-5|0.128±8.4e-5|0.118±1.0e-4|0.122±1.5e-4|0.096±4.7e-5|
>
> **R6. The improvement in written.**
>
> We will revise those sentences mentioned by the reviewers in the revised manuscript.
>
> - L38-39: Given the significant challenge of elucidating the correlation between neural activity and visual stimuli, the development of robust models for extracting vision-related latent representations is critically imperative.
>
> - L48: External latent representations aim to capture stimulus-relevant components within visual neural activity, while internal latent representations reflect dynamic internal states influencing an animal's sensory capabilities.
>
> - L117: High-quality latent representations, as compressed forms of original neural activity, must effectively reconstruct their input.
>
> **R7. Why use the contrastive loss?**
>
> Please refer to R2.
>
> **R8. Whether the resulting objective still maintains a valid ELBO?**
>
> The ELBO derivation hinges on introducing an approximate posterior for latent variables and utilizing variational inference. Contrastive learning, applied to external latent variables under the approximate posterior's input-conditioned constraint, acts as a regularizer without conflict. Its non-negative loss term also preserves the ELBO inequality.
>
> **R9. The variation in decoding performance across mice.**
>
> We have further investigated this phenomenon in Sections 4.5 and 4.6. Our findings suggest that it may stem from variations in the internal states of different mice at distinct stages, which, in turn, influence their capacity for encoding visual information.
>
> **R10. Limitations.**
>
> Without simultaneous internal state recordings, interpreting the functional roles of inferred internal latent variables is limited. Future work will analyze datasets incorporating both visual neural activity and state monitoring. Subsequently, we will extend this approach to other brain regions and experimental paradigms to investigate broader principles of biological coding mechanisms.

---

> > ### Comment · Reviewer_iXM1 · 2025-08-04
> > **Follow-up questions**
> >
> > Thanks for your explanation and additional experiments.
> >
> > **R1** Including these explanations would greatly benefit the paper. Upon reading I remembered some recent work (Schmidt, Finn, et al) that models dynamic activity of visual neurons in mouse v1 as a function of stimulus and learned latent factors, which would be appropriate to include and compare against in your manuscript. Re positioning of your work in the literature, I defer the reader to comments made by reviewer **RfuT** and your rebuttal thereof.
> >
> > **R2** Your rebuttal response on clarity preserves similar clarity issues as the main text.
> >
> > Several phrases read quite high level and hand-wavy that are difficult to interpret, e.g. "causal temporal capture", "promoting internal state spontaneity", "internal state variables enriching neural dynamics for more precise reconstruction". I would really appreciate it if responses are scientifically more precise and direct.
> >
> > Following are some specific questions:
> >
> > 1. What is the "classic VRNN"? Perhaps this is a well-established model, and it is still essential to motivate the borrowed design choices them in your manuscript.
> > 2. Could you please elaborate your following point? It would help to break it up into multiple sentences.
> > > "Finally, external RNN state variables depend only on current input and previous state, while internal state variables also depend on latent variables, reflecting how external visual stimuli influence an animal's internal states."
> > 3. Could you please elaborate what you mean by "time-dependent prior distribution exhibiting spontaneity guides the internal latent variables via KL divergence"?
> >
> > **R3**: I have seen the generation procedure for the synthetic datasets. I also understand that evaluation on synthetic is useful. My question is how the synthetic datasets that you choose to evaluate on are specifically useful in understanding your model's capabilities. Elaboration to this end is still missing. For example, do these simulated datasets comprise of stimulus-dependent as well as -independent components?
> >
> > **R4**: OK.
> >
> > **R5**:
> > 1. Encouraging to see better spike reconstruction. Have you also computed and compared the overall log likelihood of the generative model (ELBO)? Spike reconstruction gives us part of the picture.
> > 2. Thanks for acknowledging the lack of convincing evaluation of internal latent variables. Low correlation with visual stimuli is noted. How was the correlation measured, and what are the values that you observe?
> >
> >
> >
> > **R8**:
> >
> > > Contrastive learning, applied to external latent variables under the approximate posterior's input-conditioned constraint, acts as a regularizer without conflict. Its non-negative loss term also preserves the ELBO inequality.
> >
> > How? While I understand the constrative objective is non-negative, it impacts z^e values that are used in the reconstruction term. If z^e did not appear in the reconstruction term, then your objective preserves ELBO only considering z^i.
> >
> > **R9**:
> > > Our findings suggest that it may stem from variations in the internal states of different mice at distinct stages, which, in turn, influence their capacity for encoding visual information.
> >
> > Please elaborate on this as it is too high level to get a good understanding. What are your findings exactly? What distinct stages?
> >
> > **R10**:
> >
> > > Without simultaneous internal state recordings, interpreting the functional roles of inferred internal latent variables is limited.
> > What do simultaneous internal state recordings refer to exactly?
> >
> > Are there no other limitations of your approach?
> >
> > **References**:
> > Schmidt, Finn, et al. "Modeling dynamic neural activity by combining naturalistic video stimuli and stimulus-independent latent factors." arXiv preprint arXiv:2410.16136 (2024).

---

> > > ### Author Response · Authors · 2025-08-06
> > >
> > > Thank the reviewer for the insightful feedback. We will try to provide more experimental evidence and clarification on the reviewer's remaining concerns.
> > >
> > > 1). We will incorporate these explanations, the references mentioned by Reviewer RfuT, and the results of the additional comparisons into the revised manuscript. Besides, we have read the work (Schmidt, Finn, et al) before, but as its code is not publicly available, we will consider incorporating it as a baseline in our experiments once the code is made public.
> > >
> > > 2). We would like to provide more detailed clarification below.
> > >
> > > * The variational RNN (VRNN) [1] introduces an RNN and uses the RNN state variable to extend vanilla VAE for the purpose of modeling sequential data. Specifically, the latent variable $z$ at each time step depends not only on the input $x$ but also on the state variable $h$, which is updated step by step through the RNN's recurrent structure. Our model adopts these core designs of VRNN. Additionally, the concept of causal temporal modeling implies that the latent variable at the current time step depends only on the inputs from past time steps and not on future time steps. This distinguishes our model from LFADS and CEBRA, which also model temporal relationships but do not employ causal temporal modeling. LFADS uses bidirectional RNNs, while CEBRA employs temporal convolutions.
> > >
> > > * Regarding the update for the RNN state variables, external state variables $h^{(e)}$ depend solely on the input neural activity and their values from the previous time step. Differently, internal state variables $h^{(i)}$ depend not only on the input neural activity and their previous time-step values, but also on the full latent variables $z$. This design choice reflects a consideration that animals' internal states are susceptible to the influence of current external visual stimuli.
> > >
> > > * By optimizing the KL divergence between the prior distribution and the approximate posterior distribution of internal latent variables, the internal latent variables $z^{(i)}$ constructed from neural activity (approximate posterior distribution) are made closer to their spontaneous prior distribution. The spontaneity of the prior distribution refers to the internal state variables $h^{(i)}$ that construct the prior internal latent variables $\tilde{z}^{(i)}$ being autonomously recurrent (Equation 2).
> > >
> > > 3). The non-temporal dataset, being label-dependent, is designed to evaluate the model's ability to construct discriminative latent variables. Conversely, the temporal dataset assesses the model's capacity to represent temporal information. However, these two datasets do not simultaneously comprise both stimulus-dependent and -independent components. As we mentioned in our rebuttal, we evaluate the model's ability to disentangle on real datasets, but we will consider constructing more complex synthetic datasets that incorporate these different components in future work.
> > >
> > > 4). The ELBO results, presented in Tables r1 and r2, demonstrate an order of magnitude difference between the KL divergence of the latent variables and the spike reconstruction loss. Consequently, the ELBO is dominated by the spike reconstruction loss.
> > >
> > > Regarding the correlation, we reflect it through the decoding scores of internal latent variables. As shown in Figures 3A and 4A, these scores are significantly lower than those of external latent variables, which indicates a weaker correlation with visual stimuli.
> > >
> > > Table r1: The ELBO for the scene dataset.
> > >
> > > ||Mouse 1|Mouse 2|Mouse 3|Mouse 4|Mouse 5|
> > > |-|:-:|:-:|:-:|:-:|:-:|
> > > |LFADS|0.235±3.83e-05|0.222±4.51e-05|0.232±3.56e-05|0.209±5.81e-05|0.226±5.75e-05|
> > > |pi-VAE|0.254±8.71e-05|0.241±2.32e-05|0.248±2.45e-05|0.228±4.22e-05|0.245±6.23e-05|
> > > |Swap-VAE|0.203±4.16e-04|0.189±2.64e-04|0.196±2.58e-04|0.182±3.09e-03|0.201±1.09e-03|
> > > |TE-ViDS|0.195±4.89e-04|0.145±1.67e-04|0.190±5.85e-03|0.151±5.73e-03|0.189±9.69e-04|
> > >
> > > Table r2: The ELBO for the movie dataset.
> > >
> > > ||Mouse 1|Mouse 2|Mouse 3|Mouse 4|Mouse 5|
> > > |-|:-:|:-:|:-:|:-:|:-:|
> > > |LFADS|0.134±4.00e-04|0.196±5.15e-04|0.186±5.61e-04|0.192±5.41e-04|0.163±5.50e-04|
> > > |pi-VAE|0.159±7.38e-05|0.217±6.34e-05|0.205±7.84e-05|0.209±7.99e-05|0.179±6.53e-05|
> > > |Swap-VAE|0.100±2.62e-04|0.153±2.96e-04|0.144±2.45e-04|0.149±2.09e-04|0.120±2.40e-04|
> > > |TE-ViDS|0.086±4.00e-05|0.128±8.49e-05|0.118±1.03e-04|0.123±1.58e-04|0.097±2.36e-05|

---

> > > ### Author Response · Authors · 2025-08-06
> > >
> > > 5). First, it is important to note that all variables $z^{(e)}$ included in the contrastive loss are conditioned on the input neural activity (i.e., approximate posterior). Thus, while $z^{(e)}$ is regularized by the contrastive loss, its fundamental dependence on the input is maintained, ensuring it can be used for neural activity reconstruction without conflicting with the ELBO. Second, this approach of introducing additional losses to the latent variables of a VAE to achieve specific objectives is a common practice explored in some studies [2-3].
> > >
> > > 6). As detailed in lines 255-260 and further elucidated in R7 to Reviewer NU2a, our analysis of the RSM reveals distinct patterns of internal state changes across mice. We thus hypothesize that the inter-individual differences in decoding performance may be influenced by the internal state of each animal, potentially serving as a significant factor contributing to inter-individual variability.
> > >
> > > 7). Simultaneous internal state recordings encompass both physiological parameters (e.g., body temperature and heart rate) and behavioral information that may reflect an animal's internal state (e.g., pupil size and running speed).
> > >
> > > Regarding the limitations, as mentioned above, our model exhibits differences in decoding performance across different mice. A meaningful direction for future work is to develop more robust modeling methods that can effectively reduce such inter-individual variability in the latent variables.
> > >
> > > We will reflect the above-mentioned clarifications as well as the limitations of our work in the revised manuscript.
> > >
> > > [1] Junyoung Chung, Kyle Kastner, Laurent Dinh, Kratarth Goel, Aaron Courville, Yoshua Bengio. A Recurrent Latent Variable Model for Sequential Data. NeurIPS 2015.
> > >
> > > [2] Hyunjik Kim, Andriy Mnih. Disentangling by Factorising. ICML 2018.
> > >
> > > [3] Yu Wang, Hengrui Zhang, Zhiwei Liu, Liangwei Yang, Philip S. Yu. ContrastVAE: Contrastive Variational AutoEncoder for Sequential Recommendation. CIKM 2022.

---

> > > > ### Comment · Reviewer_iXM1 · 2025-08-07
> > > > **Follow-up II**
> > > >
> > > > I thank the authors for the detailed explanations. I would like to provide some more comments on some of the points. I do not have any comments or questions on the rest of the points.
> > > >
> > > > **Re point 2**
> > > >
> > > > Thanks for bringing up the reference for VRNN here directly. I would strongly recommend to include more detailed explanations of your model architecture in your manuscript, even though components are borrowed from VRNN.
> > > >
> > > > **Re point 5**
> > > >
> > > > Viewing the contrastive loss as a regularizer on $z^e$ does not, by itself, guarantee that the full objective remains a valid variational lower bound. While it is indeed common practice to augment generative models with auxiliary losses applied to their latent spaces, such additions do not inherently preserve the variational interpretation. The presence of these terms often breaks the formal ELBO unless a derivation explicitly demonstrates that they contribute to (or at least do not violate) the lower bound on the marginal likelihood. If the cited references provide such a derivation, it is important to reproduce it clearly in the manuscript, as this is a nontrivial and critical justification.

---

> > > > > ### Author Response · Authors · 2025-08-07
> > > > >
> > > > > We thank the reviewer for the feedback. As suggested, we will first include a more detailed description of our model architecture and underlying motivation in the revised manuscript. Second, ContrastVAE [3] provides the derivation of ELBO after introducing contrastive learning. Specifically, they consider the maximization of the joint distribution of positive sample pairs $\log{p(x,x_{pos})}$ and derive an additional mutual information term $I(z,z_{pos})$ beyond the vanilla ELBO by following the practice from other studies [4]. Then, the InfoNCE loss is used as a lower bound of mutual information for optimization [5]. Although our final loss function is consistent with their form, we will adopt this more rigorous derivation and restructure our loss derivation in the revised manuscript.
> > > > >
> > > > > We sincerely appreciate the reviewer's active discussions and constructive suggestions. Incorporating these suggestions into our manuscript indeed strengthens our work.
> > > > >
> > > > > [4] Laurence Aitchison. InfoNCE is a variational autoencoder. arXiv 2021.
> > > > >
> > > > > [5] Ben Poole, Sherjil Ozair, Aäron van den Oord, Alexander A. Alemi, and George Tucker. On Variational Bounds of Mutual Information. ICML 2019.

---

> > > > > > ### Comment · Reviewer_iXM1 · 2025-08-07
> > > > > > **Final remark**
> > > > > >
> > > > > > If the provided reference specifically proves that the contrastive objective maintains the variational lower bound, then it is sufficient, but still necessary to include (reproduce) in the manuscript.
> > > > > >
> > > > > > Overall, I thank the authors for their engagement and explanations. They help position their work better and strengthen their contributions.

---

### Official Review · Reviewer_RtPx · 2025-06-29

**Clarity:** 4
**Significance:** 3
**Originality:** 3
**Rating:** 4
**Confidence:** 4

**Summary:**

The paper presents a dynamical latent variable model that separates stimulus-related and internal neural dynamics using a hybrid VAE and contrastive learning framework. The model consists of two distinct latent pathways: a deterministic external latent space to capture stimulus-driven variability, and a stochastic internal latent space to capture trial-specific dynamics. A key feature is the use of contrastive learning and a swap-based reconstruction loss to enforce a functional separation between these two latent spaces. The model is evaluated on neural recordings from mouse visual cortex and the authors claim it uncovers both representational structure and hierarchical processing.

**Questions:**

Below is a list of questions, concerns, and suggestions that I had while reading the paper. Some relate to specific modeling choices and assumptions, while others concern interpretation and framing of the results:

**Questions for which I am happy to increase the rating if properly addressed:**

- What are the exact “new insights” that this paper is providing “into the intrinsic correlation between neural activity and visual stimuli” (line 54), as the authors mention in the intro? same question for “visual information processing mechanisms of the mouse visual cortex”? I agree the method has potential for being used to generate insights, but if you are claiming that it is providing the insight then the insight should be provided, rather thoroughly and convincingly. I am specifically curious about the “new” insight, as opposed to confirming previous findings (maybe even with better quality) or improved performance.
- The authors motivate the choice of making the external latent deterministic in biological terms (i.e. “to reduce the noise inherent in neural activity”). Why is that desired, from the perspective of modelling biological data? Neural responses to the same stimulus exhibit trial-to-trial variability that is not fully attributable to latent brain states or structured noise (i.e. individual neurons exhibit intrinsic variability), so a biologically faithful model would allow $\mathbf{z}^{(e)}$ to reflect some uncertainty.
- The decoder takes $\mathbf{h}_{t-1}^{(i)}$ as input in addition to the latents $\mathbf{z}_t^{(e)}$ and $\mathbf{z}_t^{(i)}$. If the latents are meant to capture the full structure of the neural dynamics, what is the justification for passing the RNN hidden state upstream of $\mathbf{z}_t^{(i)}$ into the decoder? To me, this blurs the role of $\mathbf{z}_t^{(i)}$ as a sufficient representation of the internal factors that effect high-dimensional neural data. Moreover, if the hidden state of the internal RNN is necessary, why not include the hidden state of the external RNN as well?
- How using denoised responses (i.e. using the predictions of any powerful predictive model) and computing RSM is “facilitating the unraveling of information processing mechanisms in the visual cortex”? The motivation for comparing the RSMs across two mice (Fig. 3B) was the observed difference in decoding score (i.e. mainly concerned with external factors), but the explanation is more concerned with internal states. The way I understood the explanation is that it looks like the two mice have different internal states and therefore the decoding score is different across subjects. First of all, I do not find this argument convincing, and even if I was convinced, I am not sure if this qualifies as “new insights into differences in visual processing between subjects” (mentioned in the abstract).
- Regarding the explanation of the observed diagonal structure across all video trials (Fig. 4F), I agree that this indicates the mouse neural representation is driven by stimulus-related responses—especially since internal states are expected to vary across trials. The persistence of the diagonal structure thus likely reflects the dominance of the stimulus-driven component. To support this claim more directly, it would be helpful to see RSMs computed separately on the external and internal latents. The expectation is that the RSM of the external latent should preserve the diagonal structure across trials, while this pattern should be absent in the RSM of the internal latent.
- I find the interpretation of the decoding scores across visual areas in Fig. 5 somewhat unclear. First, while the authors suggest that lower decoding in VISrl supports a hierarchical organization, they simultaneously attribute it to multi-sensory integration, which weakens the hierarchy argument. Second, if the cortex is indeed hierarchically organized, one might expect higher decoding scores in high-level areas due to more specialized and invariant representations. Instead, the highest decoding appears in VISp, the primary area, which seems counterintuitive if semantic discriminability is the goal. Could the authors elaborate how these results support a hierarchical structure, and how one might disentangle the effects of hierarchy from those of multi-modal integration?
- The claim that the results amount to a “potential breakthrough” seems overstated. While I agree that the model may offer useful insights into sources of trial-to-trial variability, their interactions, and their temporal structure, the evidence does not clearly demonstrate a substantial advancement over prior work in uncovering computational mechanisms in visual areas. I suggest rephrasing this statement to more accurately reflect the scope of the contribution.

**Other (minor) questions/suggestion:**

- Could the authors explain how using the contrastive learning objective along with the swap reconstruction loss enforces a functional separation between the two latent factor sets (i.e. $\mathbf{z}^{(i)}$ and $\mathbf{z}^{(e)}$)? What is the contribution of each of these losses?
- It would help the reader if the authors highlight the main differences between their approach and those mentioned in the related work (at least the most relevant ones that they use for model comparison). Some differences are mentioned (e.g Swap-VAE assumes independence over time) but for LFADS and CEBRA there is only a one sentence description and it is not immediately clear what the difference is (e.g. no internal-external factorization in LFADS). A more clear contrast of important aspects can help understanding the methodological contribution of the presented method.

**Ethical Concerns:**

["NO or VERY MINOR ethics concerns only"]

**Final Justification:**

Considering the authors’ responses to my concerns and their discussion with the other reviewers, I believe the merits of this work outweigh its limitations, and I have decided to maintain my positive evaluation.

**Limitations:**

Although the authors indicate in the checklist that limitations are discussed in Section 5 (Discussion), I could not find any explicit mention of limitations in the text. However, below are some limitations that I think are worth discussing (as mentioned also in my questions):

- The deterministic nature of the external latent may artificially suppress trial-to-trial variability that is genuinely stimulus-related.
- The evaluation focuses primarily on decoding and RSM structure, with limited connection to actual mechanistic hypotheses about neural computation. For instance, do the learned internal latent trajectories reflect known internal states (e.g., arousal, locomotion)? Or do the external latent encodes interpretable stimulus properties?
- The interpretation of subject differences and hierarchical processing are speculative

**Paper Formatting Concerns:**

None.

**Quality:**

3

**Strengths And Weaknesses:**

**Strengths:**

- The text is clearly written with a logical flow and was easy to follow
- Proposes a novel combination of contrastive learning, latent swapping, and variational sequence modeling.
- Explicit modeling of internal vs. external factors in neural data is conceptually appealing and clearly structured.
- The swap-based reconstruction loss is a clever mechanism for enforcing disentanglement.

**Weaknesses:**

- Several modeling choices (e.g., deterministic external latents, use of internal GRU state in the decoder) are not clearly justified given the biological modeling goals.
- Interpretations in the results sections often overreach the evidence presented (e.g., "potential breakthrough") and claims about “new” insights into neural processing mechanisms and hierarchical structure are not strongly supported by the presented analyses.

---

> ### Author Rebuttal · Authors · 2025-07-31
>
> We thank the reviewer for the thoughtful and detailed comments. We will do our best to address the comments and provide detailed responses point by point, below.
>
> **R1. About new insights.**
>
> Our work offers two key insights into visual information processing mechanisms in the mouse visual cortex. First, the encoding capabilities for visual information are influenced by the animals' internal state, potentially accounting for variability in neural activity both across individuals and within the same individual over time. Second, under long-duration continuous visual stimuli, the neural activity of mice becomes more focused on encoding visual stimuli and is less influenced by their internal state. Further details on these points are elaborated in R4 and R5.
>
> We acknowledge, however, that certain phrasing, such as claims of providing "new insights" into the intrinsic correlation between neural activity and visual stimuli, was imprecise. This should be revised to state that our model captures a greater degree of correlation between neural activity and visual stimuli. We will implement these necessary modifications in the revised manuscript.
>
> **R2. About the deterministic external latent variables.**
>
> We agree that factors beyond neuronal noise and internal brain states contribute to trial-to-trial variability. However, spontaneous neural activity, which is highly correlated with internal states, is recognized as a predominant source of cortical variability [1].
>
> To comprehensively model variability involving external inputs, it would be necessary to incorporate auxiliary information, such as neural activity from other cortical regions (e.g., encoding other sensory inputs, behaviors, or memories). Given that this work analyzes neural activity within visual regions, an attempt to rashly model such externally driven variability (e.g., by pre-specifying the variance of external latent variables) risks both an ineffective representation of this variability and a detrimental impact on the modeling of internal state-related latent variables.
>
> Therefore, to maintain our focus on variability arising from internal brain states, we have opted to simplify the modeling of external latent variables as deterministic values.
>
> **R3. About the inputs of decoder.**
>
> We admit that the choice of state variables as input to the decoder is primarily motivated by engineering considerations. First, following the architectural dependencies established in the VRNN, we introduce internal state variables in the decoder to further enrich the dynamic information, extending beyond what is captured by internal latent variables, thereby facilitating a more accurate reconstruction of neural activity. Second, since the external latent variables are deterministic functions of external state variables, introducing external state variables into the decoder yields less information gain. Therefore, we omitted this inclusion to mitigate computational complexity.
>
> **R4. How computing RSM is "facilitating the unraveling of information processing mechanisms in the visual cortex"?**
>
> External latent variables reflect neural activity components related to visual stimuli; consequently, their performance in decoding these stimuli cannot surpass the inherent validity of the visual information encoded by the neural activity itself. Therefore, observed differences in decoding scores across individual mice are more plausibly attributed to differences in the visual information encoded by their respective visual neural activity, rather than being a direct consequence of the external latent variables. More abstractly, these variations may be linked to the visual sensory capabilities of different mice at various times.
>
> Building upon this premise, our analysis of the RSM reveals distinct patterns of internal state changes (as reflected by the internal latent variables) across mice, as detailed in lines 255-260 and further elucidated in R7 to Reviewer NU2a. We thus hypothesize that such inter-individual differences in sensory performance may be influenced by the internal state of each animal, potentially serving as a significant factor contributing to inter-individual variability. This investigation into the influence of internal states on sensory performance supports our broader aim of exploring and explaining the mechanisms of visual information processing across different individuals.
>
> **R5. About the Figure 4F.**
>
> As suggested by the reviewer, we compute the RSM for external latent variables and internal latent variables separately. As expected, the RSM for external latent variables consistently exhibits a clear diagonal structure, even for different trials. In contrast, the RSM for internal latent variables lacks such a structure, displaying no consistent organization across trials. These findings further support the hypothesized distinct roles of the external and internal latent variables in encoding different aspects of neural activity. We will incorporate these additional results into the revised manuscript.
>
> **R6. The interpretation of the decoding scores across areas.**
>
> First, our conclusions regarding hierarchical organization are primarily drawn from observed differences in decoding performance between primary and high-level cortical regions (lines 310-313). We wish to clarify that the discussion concerning the lowest decoding performance in VISrl (a multisensory brain region) serves merely as a potential explanation for this specific observation, rather than as an argument for hierarchical organization. Second, we offer the following explanation for the superior decoding performance observed in VISp. The neural datasets used in our experiments include categories with similar semantic content (e.g., various natural scenes featuring the same animals, or movie frames that are temporally distant but depict similar characters). Consequently, high-level cortical regions (VISpm and VISam), which primarily encode high-level visual features such as semantic information, may struggle to distinguish these subtle categorical differences. Conversely, the primary visual cortex (VISp), which encodes low-level visual features like texture and orientation, is better equipped to discriminate these fine-grained distinctions.
>
> Nevertheless, we acknowledge that our initial conclusions regarding functional hierarchy were overly strong. The current results primarily suggest that the visually encoded information is heterogeneous across different mouse cortical regions. We will temper our conclusions accordingly in the revised manuscript. Furthermore, the interactive effects of hierarchical structure and multimodal integration represent a compelling avenue for future exploration, likely necessitating a synergistic analysis of neural activity across multiple sensory cortices.
>
> **R7. About the potential breakthrough.**
>
> It appears there may be a misunderstanding regarding the "potential breakthrough" referenced in lines 330-331. We wish to clarify that this phrase pertains to a promising direction for future research—specifically, the challenge of constructing stable, high-quality latent representations across diverse experimental conditions. It is not presented as a contribution of our work. In fact, the observed variability in our model's decoding performance across different mice is a recognized limitation of this work, underscoring the need for further investigation into robust modeling.
>
> **R8. What is the contribution of each of these losses?**
>
> Our approach achieves disentanglement between external and internal latent variables through multiple training objectives.
>
> First, as described in lines 132–141, contrastive learning is applied to the external latent variables. This mechanism encourages external latent variables corresponding to sample pairs with small temporal offsets (i.e., under similar visual stimuli) to cluster together, thereby establishing an association between these variables and the visual input. Concurrently, a time-dependent prior distribution with spontaneity is used to guide the internal latent variables via the KL divergence. The application of these distinct loss functions to the two types of latent variables is instrumental in promoting their disentanglement.
>
> Second, the swapping operation (details in R1 to Reviewer NU2a) further reinforces this disentanglement. This operation involves exchanging the external latent variables between positive sample pairs while preserving their internal latent variables. This manipulation specifically aims to further reduce the distance between the external latent variables of positive pairs. By doing so, it compels the model to isolate and strengthen the distinct roles of the external and internal latent variables, thereby enhancing their disentangled representation.
>
> **R9. The main differences between our model and alternative models.**
>
> We have provided a brief description of the differences in Section 4.3. Here, we elaborate on the main differences between our proposed model and alternatives:
>
> * pi-VAE: Unlike our self-supervised approach, it incorporates additional supervised information (specifically, visual stimulus categories) for latent variable construction. Additionally, pi-VAE does not model the temporal dynamics inherent in neural activity.
>
> * Swap-VAE: Similar to pi-VAE, Swap-VAE does not explicitly model the temporal information of neural activity, operating under an assumption of temporal independence.
>
> * LFADS and CEBRA: A key distinguishing feature is that neither LFADS nor CEBRA performs a decomposition into external and internal latent variables. Moreover, the temporal relationships modeled by these approaches are not strictly causal. The former employs bidirectional RNNs to capture temporal relationships, while the latter uses temporal convolutional kernels to extract time information.
>
> [1] Spontaneous behaviors drive multidimensional, brainwide activity. Carsen Stringer, et al. Science 2019.

---

> > ### Comment · Reviewer_RtPx · 2025-08-04
> > **Response to Author Rebuttal**
> >
> > I thank the authors for their explanation and additional analysis. Some of my concerns were addressed, but some still remain.
> >
> > The decision to include the internal state variable in the decoder is described as an engineering choice to improve reconstruction. Given the goal of achieving a meaningful internal-external latent factorization, this design choice feels somewhat under-motivated and blurs the intended separation.
> >
> > Regarding R4, I do not find the explanation convincing. If differences in decoding performance across mice are attributed to variability in how much stimulus-related signal is encoded in the data, how does that lead to the conclusion that internal states are responsible? This leap isn’t clearly justified. For example, if a subject closes their eyes and decoding accuracy drops, is that best explained by a change in internal state or simply by the absence of visual input? It would also help to clarify whether the internal latents relate in any way to known brain state variables. If so, it would be useful to demonstrate this; if not, the interpretability claim becomes harder to support.
> >
> > The authors also note in their response that “the observed variability in our model's decoding performance across different mice is a recognized limitation of this work”. This, among other limitations, should be mentioned in the paper. The lack of mention or discussion of limitations was also noted by other reviewers. I would strongly encourage the authors to explicitly mention and briefly discuss key limitations in the revised manuscript.
> >
> > Overall, I continue to view the paper positively and think the proposed approach has potential. However, given the remaining concerns and the points raised by other reviewers, I’m inclined to keep my current score for now, though I may revisit it depending on how the broader discussion evolves.

---

> > > ### Author Response · Authors · 2025-08-06
> > >
> > > Thank the reviewer for the feedback. We will provide more clarification on the reviewer's remaining concerns.
> > >
> > > 1). This design is indeed worth considering. We will try to find a balance between reconstruction and factorization.
> > >
> > > 2). As shown in Figure 3B and additional RSMs of internal latent variables, a regular, long-term (spanning multiple trials) pattern in the internal latent variables is observed in Mouse 1, and most models yields a significantly higher decoding score on this mouse. This pattern is absent in Mouse 2 and other subjects. Therefore, we hypothesize that internal states are a plausible factor influencing visual encoding information, while acknowledging that other contributing factors exist.
> > >
> > > Additionally, as discussed with other reviewers, the current work does indeed have limitations in that it is unable to quantitatively measure the relationship between internal latent variables and real internal states. We appreciate the reviewers' collective suggestions and will include a discussion of these limitations in the revised manuscript.

---

> > > > ### Comment · Reviewer_RtPx · 2025-08-06
> > > > **Clarification on the Role of Internal Latents in Decoding Performance**
> > > >
> > > > You mentioned that a "regular, long-term (spanning multiple trials) pattern in the internal latent variables is observed in Mouse 1", which is absent in Mouse 2; and since most models yield a higher decoding score for Mouse 1 than for other mice, you "hypothesize that internal states are a plausible factor influencing visual encoding." In summary, you suggest that a latent factor — captured by the internal latent variables — underlies the higher decoding accuracy observed in Mouse 1.
> > > >
> > > > But if this factor resides in the internal latents and is absent from the external latents, then one would expect the external latent decoding to be worse than that of the full model, since it would be missing this critical factor which underlies higher decoding accuracy.
> > > >
> > > > Yet Figure 3A shows the opposite: the external latent alone achieves even higher decoding accuracy than the full model. How do you reconcile this contradiction?

---

> > > > > ### Author Response · Authors · 2025-08-07
> > > > >
> > > > > We appreciate the opportunity to clarify a potential misunderstanding regarding our previous response. As stated in our original rebuttals R1 and R4, our hypothesis is that internal states may influence **the visual cortex's capability** to encode visual information, thereby leading to differences in visual neural representations (i.e., components of neural activity related to visual stimuli). In other words, although internal states may influence visual encoding capability, **the specific visual information encoded in neural activity is captured by external latent variables**. Therefore, it is reasonable for the decoding scores of the external latent variables to be higher than those of the full latent variables.
> > > > >
> > > > > We thank the reviewer for their critical engagement with this core conclusion, which has provided us with an opportunity to articulate our findings more clearly. We will refine the wording in our revised manuscript to more accurately reflect these findings.

---

> > > > > > ### Comment · Reviewer_RtPx · 2025-08-07
> > > > > > **Concluding Comment on the Correspondence between Internal Factors and Decoding Performance**
> > > > > >
> > > > > > Thank you for the further clarification. If my understanding is correct: based on Fig. 3, you are saying there seem to be internal factors that affect the overall decoding capability. But whatever the decoding capability is, it is captured by your external latent variables. And this is supported by the observation that:
> > > > > > 1. Mouse 1 decoding score is higher than Mouse 2, and the RSM in Mouse 1 shows a stronger low-dimensional structure compared to Mouse 2
> > > > > > 2. The decoding performance based on the external latent variables is high (similar level as the complete model as expected)
> > > > > >
> > > > > > First of all, this claim that "internal states may influence the visual cortex's capability to encode visual information" needs to be quantified. Specifically, a figure that shows a correspondence between the prominence of such low-d structure in RSM and the decoding score, and how strongly it is captured by the internal latent variables. A strong correspondence between these would support the authors’ claim. Comparing two mice qualitatively is certainly not enough.
> > > > > >
> > > > > > Additionally, while I believe such an analysis is essential to substantiate the claim, I would also point out that even if the claim is properly and quantifiably shown, it would not be a new finding (we know that internal factors like attention affect encoding) but rather a proof of concept to show that the model yields previously known results.
> > > > > >
> > > > > > Given the latter point and also considering that the rebuttal period is coming to an end, I am of course not asking for any additional analysis within the rebuttal period. But I would highly encourage the authors to:
> > > > > >
> > > > > > - conduct such analysis and provide clear evidence that supports the correspondence between inter-individual decoding score variability and the prominence of such low-dimensional structure (and how well it is captured by the internal latent variables)
> > > > > > - revise the claims and align them better with the actual evidence and analysis done in the paper

---

> > > > > > > ### Author Response · Authors · 2025-08-07
> > > > > > >
> > > > > > > We thank the reviewer for the feedback. In response to Reviewer NU2a, we conducted a simple quantitative analysis for the correspondence between the prominence of low-dimensional structure in the RSM and the decoding score. After computing the RSM of the internal latent variables, we observed a more pronounced regular pattern in Mouse 1. We then quantitatively assessed the similarity between two blocks of trials using the Pearson correlation coefficient. The results show that the similarity for Mouse 1 (0.248) is significantly higher than that for Mouse 2 (0.022), which suggests a correspondence between the observed structure and decoding score to some extent.
> > > > > > > While these findings are preliminary, we agree that a more rigorous metric analysis is needed to fully elucidate this correspondence, which would likely require real biological internal state data. Additionally, we will temper our claims in the revised manuscript to align with the current results, as suggested by the reviewer.

---

> > > > > > > > ### Comment · Reviewer_RtPx · 2025-08-07
> > > > > > > > **Final Remark**
> > > > > > > >
> > > > > > > > Yes, that is a good complementary analysis that, if I understood it correctly, would address my suggestion as well, **but it's important to apply this to all mice** and not just two of them. In other words, one needs to show whether this observation is consistent across all subjects.
> > > > > > > >
> > > > > > > > To conclude our discussion, I thank the authors for their engagement during the discussion period. I maintain my positive evaluation of the work.

---

### Official Review · Reviewer_NU2a · 2025-07-03

**Clarity:** 3
**Significance:** 2
**Originality:** 2
**Rating:** 5
**Confidence:** 4

**Summary:**

The authors:
- Introduce a new latent variable model (LVM) which employs two RNNs to encode spike trains from mouse visual cortex into two latent variable vectors
- The first, ‘external’ latent variable is a deterministic representation of stimulus-dependent neural activity, while the second, ‘internal’ latent variable stochastically encodes the noisy component of the neural activity
- The networks and their link functions are jointly trained to maximise the ELBO of the neural data under a time-dependent Possion generative model and a contrastive loss on the external latent variables
- The positive examples for contrastive loss in the external latent space are selected by encoding a temporally offset input sequence; negative samples are randomly selected from the dataset

**Questions:**

- Please could the authors clarify, e.g. in the appendix, how this method was applied to non-temporal synthetic data? Namely, how was sequential data generated from the stationary latent variables, thereby allowing application of the contrastive loss?
- An ablation is performed where the contrastive loss is removed. Did the authors consider another ablation where the negative examples are instead drawn from the same sequence but with longer time delays, akin to how negative samples are drawn for CEBRA? This may help isolate the importance of the chosen form of contrastive loss.
- In section 4.5, the authors claim that qualitative differences in the RSA for mice 1 and 2 are because “sensory performance in [mouse 1] is strongly influenced by their internal state.” Do the authors expect that this internal state is represented in some way by their internal latent variable? For example, could we expect similarity in the internal latents in the two blocks of trials for mouse 1, but not for mouse 2? This would help illuminate the architectural choices of the model too.
- Why does the decoding score concatenate vectors in the scenes case, but average them in the movies case?

**Ethical Concerns:**

["NO or VERY MINOR ethics concerns only"]

**Final Justification:**

As per my response to rebuttal 1, the authors intend to extend their results to fill the gaps I identified in my review. As such, I am strengthening my recommendation for acceptance.

**Limitations:**

There are no relevant negative societal impacts associated with this work.

Limitations of the model are not discussed to great extent, nor are limitations of the analysis provided.

The authors focus on one neural dataset, so a small acknowledgement of the feasibility of application to other datasets, e.g. other mouse brain areas, other species, and other tasks, would help contextualise the study.

**Quality:**

3

**Strengths And Weaknesses:**

Strengths
- This is a very interesting, novel LVM, and the authors provide a convincing set of results comparing their model to existing LVMs, both with and without time dependence.
- Furthermore, they provide an extensive range of ablation studies to their method, indicating the importance of each component of their full model. They also show that their model’s success is not due to its complexity, given that a smaller number of parameters also achieves state of the art performance on real and synthetic datasets.
- Finally, they go beyond LVM performance metrics, and uncover representational similarities of neural representations which are not evident in the raw neural data.


Weaknesses
- Architectural choice wasn’t always clear to me. While I appreciate the division of representation between stimulus and internal noise, specific dependencies in section 3.1 were not fully motivated, and authors less familiar with existing cited models could benefit from an explanation of each component. Other explanations lack some clarity, for example, details of the swapping operation and the intuition for its benefit to training.
- Without extra details, potentially provided in cited works, it is not obvious to me how the model was applied to non-temporal data in the first place. Both the generative model and the form of the contrastive loss function require a sequence of activity,
In section 4.4 it is stated that “Swap-VAE and CEBRA [suffer] a slight degradation due to the use of time-jittered positive samples.” Did the authors mean SWAP-VAE and pi-VAE - as it appears from Fig. 2E that CEBRA suffers a drastic decrease in performance due to time shuffling.
- In section 4.6, the authors claim that “our model achieves significantly higher 273 decoding scores than LFADS and is slightly better than the others” - it would be helpful to provide details on statistical tests they performed to qualify this statement.
In its current form, I find the claim that the model “provides new evidence for the functional hierarchy of the mouse visual cortex” too strong. The corresponding section 4.7 simply provides a variety of performance metrics for various parts of mouse visual cortex, but does not provide sufficient evidence to support a functional hierarchy. Such a claim would require, e.g., equivalent analysis for multisensory stimulus exposure experiments.
- Alternatively, finding a way to cross-validate the model trained one brain region in encoding activity from another region may help reveal functional dependencies between them. One method could be to train the model on one region, then retrain only the encoder and decoder weights on a second brain region, to ascertain whether the regions share similar time-dependencies.

---

> ### Author Rebuttal · Authors · 2025-07-31
>
> We thank the reviewer for the constructive and detailed comments. We will do our best to address the comments and answer the questions. Below are our detailed responses.
>
> **R1. Architectural choice and swapping operation.**
>
> Our model leverages the architecture of sequential VAEs, with many component dependencies derived from the classic model VRNN. Specifically, in the encoder, the external latent variables $z^{(e)}$ and internal latent variables $z^{(i)}$ are conditioned on the input neural activity, as well as the respective state variables ($h^{(e)}$ and $h^{(i)}$) of two distinct RNNs. This conditioning facilitates the capture of temporal information in a causal order. The prior distribution of $z^{(i)}$ is conditioned solely on the state variables. In the decoder, the reconstructed neural activity is conditioned on the $z$ and $h^{(i)}$. The deliberate inclusion of $h^{(i)}$ here is intended to enrich the information of neural dynamics, providing a more precise reconstruction. Finally, regarding the update for the RNN state variables, $h^{(e)}$ depend solely on the input neural activity and their values from the previous time step. Differently, $h^{(i)}$ depend not only on the input neural activity and their previous time-step values, but also on the latent variables. This design choice reflects a consideration that animals' internal states are susceptible to the influence of current external visual stimuli.
>
> The swapping operation for details is as follows: For a given sample and its corresponding positive sample, we exchange their external latent variables while keeping their internal latent variables unchanged. This process yields two new latent variables. We then decode neural firing rates using these new latent variables and compute the additional reconstruction loss. The intuition is this: our model is trained to reduce the distance between the external latent variables of positive sample pairs. Consequently, if the model has learned an effective external latent space, it should still accurately reconstruct neural activity even after this swap. This robust reconstruction, even with interchanged external latent variables from positive pairs, significantly enhances the training process by imposing strong regularization and encouraging more meaningful latent representations.
>
> We will include the necessary clarifications in our revised version.
>
> **R2. How the model was applied to the non-temporal dataset.**
>
> For the model input of non-temporal datasets, each sample can be conceptualized as a sequence comprising a single time step (as described in Appendix F). In the context of contrastive learning for such data, the approach of defining positive pairs through temporal shifting is inapplicable. Instead, we adopt a strategy provided in CEBRA, selecting positive samples based on the distance of their respective labels (Appendix C), thereby implementing a form of supervised contrastive learning.
>
> **R3. The sentence on line 232.**
>
> We apologize for the imprecision in the original statement on line 232. Our intended meaning was that, while both our model and LFADS exhibited substantial performance degradation, the performance reduction observed in Swap-VAE and CEBRA was comparatively less severe (although CEBRA's decline was still notable). We will revise the sentence to clarify this distinction.
>
> **R4. Statistical tests for the sentence on line 272.**
>
> Following the reviewer's suggestions, we conducted a one-sided t-test comparing the performance of our model against baseline models, with the alternative hypothesis stating that our model's mean performance is superior. The results (t-statistic > 10; p-value < 1e-5) demonstrate that our model consistently exhibits significantly higher performance than all other models across nearly all mice, with the sole exception being the comparison with CEBRA in Mouse 3. The intent of the description on line 272 is to emphasize the differential magnitude of performance gains, specifically, that our model's advantage over LFADS is greater than that over the other models.
>
> **R5. The experiment for mouse cortical regions.**
>
> In the current results (Figure 5), performance variations across brain regions, under differing visual stimuli, exhibit similar patterns. This at least indicates heterogeneous and specialized visual encoding information across each brain region. However, we acknowledge that the initial conclusion regarding a sequential functional hierarchy was overly strong, and we will temper this statement in the revised version.
>
> Following the reviewer's suggestion, we fine-tuned a model pre-trained on one region to another. This involves training only the first encoder layer and the last decoder layer, while freezing all other parameters. Tables r1 and r2 present these results, with rows indicating the pre-trained region and columns the fine-tuned region. We observe that even when pre-trained in different brain regions, the performance of the fine-tuned model shows no significant difference. The performance is lower than that of a model trained directly on the target region. This phenomenon suggests that neural activity across different brain regions may share similar latent spaces but encode distinct visual information.
>
> Table r1: The fine-tuned results (\%) for decoding natural scenes.
>
> ||VISp|VISl|VISrl|VISal|VISpm|VISam|
> |-|:-:|:-:|:-:|:-:|:-:|:-:|
> |VISp|39.68±1.83|18.88±1.83|4.00±0.18|24.00±1.30|12.95±1.26|9.08±0.80|
> |VISl|25.76±0.60|20.07±1.15|3.22±0.44|23.93±1.49|9.46±0.79|7.46±0.77|
> |VISrl|23.73±0.90|15.29±1.13|5.64±0.31|20.00±1.53|9.69±0.84|7.69±0.56|
> |VISal|25.76±0.67|17.53±1.79|4.64±0.48|27.97±1.23|12.27±1.87|8.24±0.54|
> |VISpm|25.05±1.74|14.00±0.73|3.59±0.45|19.53±1.54|13.42±1.00|7.66±0.83|
> |VISam|24.37±1.84|16.31±1.47|4.75±0.27|22.68±1.02|11.29±0.48|10.22±0.53|
>
> Table r2: The fine-tuned results (\%) for decoding natural movie frames.
>
> ||VISp|VISl|VISrl|VISal|VISpm|VISam|
> |-|:-:|:-:|:-:|:-:|:-:|:-:|
> |VISp|35.33±1.79|24.44±0.76|15.33±0.84|29.84±1.87|22.60±0.88|23.56±1.07|
> |VISl|28.71±1.63|28.71±0.88|13.96±0.74|31.53±1.21|20.82±0.74|22.80±0.97|
> |VISrl|28.62±1.56|24.44±0.49|15.78±0.61|30.44±1.27|21.73±0.75|23.33±0.70|
> |VISal|28.64±1.52|24.44±0.91|14.60±0.72|38.80±1.97|20.07±0.64|25.09±1.56|
> |VISpm|28.73±1.54|24.09±0.62|14.00±0.34|31.09±1.13|25.07±1.51|23.04±0.96|
> |VISam|31.18±1.72|23.51±0.96|14.24±0.25|32.80±1.17|21.02±1.08|27.20±0.91|
>
> **R6. Ablation study about the negative samples.**
>
> In response to the reviewer's suggestions, we investigate an alternative negative sampling strategy: drawing negative samples from the same sequence but with longer time offsets. Specifically, for the scene dataset, we use a 100 ms offset for negative samples (compared to 30 ms for positive samples). For the movie dataset, negative samples are generated with offsets of approximately 330 ms and 165 ms (versus an approximate 16 ms offset for positive samples). As detailed in Tables r3 and r4, this negative sampling approach results in a slight performance decline, with a larger decrease observed when the negative sample offset is smaller. These results indicate the feasibility of this alternative strategy, yet they also highlight a trade-off: selecting appropriate offsets for different datasets increases the computational cost associated with hyperparameter search. Our current methodology, which involves randomly selecting negative samples from the entire training dataset, emerges as a more generalized and efficient approach. Besides, we carefully review the code of CEBRA, confirming that it adopts the same random sampling strategy.
>
> Table r3: The ablation results (\%) for decoding natural scenes.
>
> ||Mouse 1|Mouse 2|Mouse 3|Mouse 4|Mouse 5|
> |-|:-:|:-:|:-:|:-:|:-:|
> |Random|50.86±0.81|27.24±0.47|29.90±0.43|38.05±0.53|9.44±0.20|
> |Longer offset (100 ms)|48.20±0.48|26.46±0.52|27.49±0.32|36.47±0.39|8.31±0.23|
>
> Table r4: The ablation results (\%) for decoding natural movie frames.
>
> ||Mouse 1|Mouse 2|Mouse 3|Mouse 4|Mouse 5|
> |-|:-:|:-:|:-:|:-:|:-:|
> |Random|13.88±0.19|65.38±0.36|59.88±0.72|54.33±0.54|30.18±0.40|
> |Longer offset (300 ms)|13.00±0.31|63.96±0.47|59.37±0.84|53.78±0.45|29.37±0.54|
> |Longer offset (165 ms)|12.83±0.29|63.24±0.70|58.58±0.41|52.40±0.53|28.62±0.49|
>
> **R7. Do the authors expect that this internal state is represented by their internal latent variable?**
>
> In fact, the RSM of mouse 1 in Figure 3B has already shown regular changes in the internal state, i.e., the two redder blocks in the top-left and bottom-right corners, as discussed in lines 255–258 of the text. Upon checking the RSM of the internal latent variables, we observe that this pattern is even more pronounced. Subsequently, a quantitative assessment of the similarity between two blocks of trials, performed using the Pearson correlation coefficient, reveals that the similarity for Mouse 1 (0.248) is much higher than that for Mouse 2 (0.022). This finding provides further corroboration for our statement that visual sensory performance is influenced by internal states.
>
> **R8. Why does the decoding score concatenate vectors in the scenes, but average them in the movies?**
>
> The presentation duration of static scenes (250 ms) necessitates the preservation of dynamic changes within their latent representations for effective stimulus discrimination. Consequently, a concatenation operation is employed to retain maximal temporal information. In contrast, the frame rate of the movie is 30 Hz, meaning that each movie frame lasts only about 33 ms. Using an averaging operation for the latent representation of a single movie frame does not result in much information loss but can reduce the computational complexity of KNN algorithms.
>
> **R9. Limitations.**
>
> The main limitation is the inability to quantitatively analyze internal latent variables reflecting internal states. Moreover, as noted in lines 332-334, extending our method to other brain regions and species represents a promising avenue for future exploration.

---

> > ### Comment · Reviewer_NU2a · 2025-08-02
> >
> > I thank the authors for both their conceptual clarifications and the extensive extensions to their empirical results. With these gaps filled, I believe the paper's results are in stronger shape. As such, I will strengthen my recommendation for acceptance.

---

> > > ### Author Response · Authors · 2025-08-02
> > >
> > > Thank you very much. We are encouraged by your recognition of our work and will ensure that the improvements we've discussed are thoroughly incorporated into the revised manuscript.

---

### Note · Authors · 2025-08-12

We are pleased to have this opportunity to summarize the discussion on our work. We will emphasize the key clarifications and planned revisions below.

1. **Model Positioning**. To clarify our model's position and contributions for reviewer iXM1, we addressed its scientific objectives and methodological differences from existing work. Furthermore, as requested by reviewer RfuT, we have compared our model with a broader range of baselines and confirmed the superiority of our model. We will add a broader discussion of relevant literature and these new comparison results to the revised manuscript.

2. **Model Design**. In response to the common comments from reviewers NU2a, RtPx, and iXM1, we have provided more detailed descriptions of the design and motivation of our model's architecture and loss functions. This content will be incorporated into the revised manuscript.

3. **Additional Evaluation**. As suggested by reviewer iXM1, we have conducted additional evaluations on spike reconstruction and the overall log likelihood (ELBO). The results consistently show that our model outperforms the baselines on these metrics.

4. **Scientific Conclusion**. In response to the concerns raised by reviewers NU2a and RtPx, we have clarified the scientific conclusions drawn from Figures 3 and 5 and offered additional evidence. Nevertheless, we acknowledge that some of our initial statements were overly strong and will temper these claims to align with the current results.

5. **Limitation**. As requested by all reviewers, we have re-examined the limitations of our work, including the inability to quantitatively analyze internal latent variables reflecting internal states and the variability in decoding performance across different mice. Relevant discussions of these limitations will be included in the revised manuscript.

In summary, our responses have addressed most of the reviewers' concerns. While some disagreements remain, we believe that our work represents a notable contribution to the application of latent variable models in visual neural activity research. We sincerely appreciate all the thoughtful and meaningful comments, which have been instrumental in improving our work.

---

### Decision · Program_Chairs · 2025-09-17

**Decision:**

Accept (poster)

**Comment:**

This paper introduces a time-dependent latent variable model that incorporates separate componentns for stimulus and noise, and is trained using a contrastive learning framework.  Its strengths include a novel modeling approach to an important problem in neuroscience (i.e., resolving stimulus and noise underlying neural responses) and is well written.  Weaknesses involved lack of justification for some modeling choices and inadequate comparisons to other baseline methods.  Ultimately, I found the authors' rebuttal responses compelling and feel that the reservations of reviewer RfuT are not fatal to the paper's contributions.  (I agree, for example, that some of the alternate methods -- eg that used supervised approaches to identifying signal latents, or methods focused on low-D motor variables -- are not entirely relevant benchmarks to the unsupervised method proposed here).  I thus feel the paper makes a valuable contribution to the NeurIPS community and am happy to recommend it for acceptance.  Please be sure to address all reviewer comments and criticisms in the final manuscript.